# Neuronal populations across the cortex underlie discrete, categorical, and subjective representations of visual durations

Valeria Centanino<sup></sup>, Gianfranco Fortunato<sup></sup>, Domenica Bueti

International School for Advanced Studies (SISSA), Trieste, Italy

☯ These authors contributed equally to this work.
* vcentani@sissa.it (VC); gfortuna@sissa.it (GF)

## Abstract

The neural processing of subsecond durations recruits a wide network of areas. Although unimodal tuning has been shown in many of these regions, its role and link to perception remain unclear. Here, we used 7T functional MRI while participants performed a visual duration categorization task to characterize unimodal responses along the cortical hierarchy. We found topographically organized neuronal populations tuned to all presented durations in parietal and premotor cortices, and in the caudal supplementary motor area (SMA). In contrast, rostral SMA, inferior frontal cortex, and anterior insula showed neuronal preferences centered around the mean duration, which correlated with the boundary duration participants employed in the task. These differences suggest specialized roles of duration tuning across cortical regions —from discrete to categorical and subjective duration representations. Finally, correlations of neuronal preferences across areas highlighted a hierarchical organization of duration tuning. Together, our findings provide a mechanistic framework for duration perception in vision.

## Introduction

Time is a pervasive dimension of our everyday experience, and our continuous interaction with the environment relies on our ability to track and anticipate events unfolding within hundreds of milliseconds. In the past decades, a significant body of research in humans has shown that the processing of millisecond time engages several cortical and subcortical brain areas. Most of these brain regions seem engaged in temporal computations across a variety of tasks, sensory modalities, and temporal ranges [1–3]. Beside the identification of this "timing network," animal electrophysiology [4–10] and human functional magnetic resonance imaging (fMRI) [11–15] studies indicate that the brain processes millisecond time through neuronal units tuned to specific durations (i.e., exhibiting a unimodal response function). In humans, these neuronal units, which have been identified across a wide cortical network

**Data availability statement:** The data that support the findings of this study are available on OSF at the following link: https://doi.org/10.17605/OSF.IO/2TEQU. MRI data are provided in a preprocessed format to comply with privacy regulations.

**Funding:** This project has received funding from the European Research Council (ERC, https://erc.europa.eu/homepage) under the European Union's Horizon 2020 research and innovation programme (grant agreement no. 682117 BIT-ERC-2015-CoG) to D.B., and from the Italian Ministry of University and Research (https://www.mur.gov.it) under the call PRIN22 (project ID: 2022CCPJ3J) to D.B. The funders did not play any role in the study design, data collection and analysis, decision to publish, or preparation of the manuscript.

**Competing interests:** The authors have declared that no competing interests exist.

**Abbreviations:** CSF, cerebrospinal fluid; EPI, echo-planar imaging; FD, framewise displacement; fMRI, functional magnetic resonance imaging; GLM, general linear model; GM, gray matter; HRF, hemodynamic response function; INU, intensity non-uniformity; pRF, population receptive field; PSE, point of subjective equality; ROIs, regions of interest; SCI, stimulus-cue interval; SMA, supplementary motor area; TE, echo time; TR, repetition time; WM, white matter.

spanning lateral occipital, parietal, and frontal areas, are topographically organized in maps (i.e., neuronal units with similar duration preferences are spatially contiguous on the cortical surface) [11–15]. Interestingly, there is some overlap between the areas where temporal maps ("chronomaps") have been observed and those previously identified as part of the timing network. However, the functional relationship between duration-tuned neural populations and timing areas remains elusive. In the first place, not all the areas of the timing network show unimodal tuning. For example, unimodal tuning has never been documented in the anterior insula, which is consistently activated by timing tasks [1–3] and considered central to the embodied account of time [1,3,16–18]. Second, unimodal tuning is not consistently reported in some brain areas. For instance, in the supplementary motor area (SMA), an area widely recognized for its central role in temporal computations [2,3,19] and where unimodal responses to stimulus durations have been recorded in monkeys [4,5,20], chronomaps are not always observed in humans [12–14]. Third, the redundancy of chronomaps across brain areas and their functional contribution to time processing and perception remains unclear. Evidence supporting a structured cortical organization of duration tuning, that also reflects functional differentiation, comes from two recent fMRI studies that focused on the visuo-spatial hierarchy [11,15]. These studies showed that unimodal responses to duration gradually emerge from extrastriate visual areas onward, while at the earliest stages of visual processing, responses to durations are monotonic. This cortical organization, characterized by a shift from monotonic to unimodal duration tuning, suggests a transition in temporal processing: from an initial stage rooted in visual coding, where temporal information is extracted from the incoming event, to a subsequent stage dedicated to reading out that information. However, it remains unclear whether further changes in duration tuning occur along the cortical hierarchy to support other stages of temporal processing and, ultimately, shape duration perception. An indirect measure of the link between duration tuning and perception was reported by Hayashi and Ivry, who combined a duration adaptation paradigm with fMRI [21]. The authors found that, in the right supramarginal gyrus, durations similar to the adaptor stimulus evoked an attenuated response, consistent with the presence of duration-tuned populations that become suppressed following adaptation. Importantly, this modulation of brain response predicted the behavioral bias, supporting a connection between duration-tuned responses and perception. Further evidence for this link comes from studies showing a task dependence of duration-tuned populations in SMA. Duration maps in this area partially reshape when the tested duration range changes [12] or are not observed when participants were not actively engaged in a task [13,14]. This latter finding is consistent with observations in other types of neural maps, whose presence depends on task demands. For instance, retinotopic maps in frontoparietal cortices are systematically and robustly elicited only when visual field mapping is combined with cognitively demanding tasks engaging attention or memory [22–24]. Overall, while existing evidence suggests a degree of dependence between duration-tuned populations and timing behavior, their direct relationship remains to be explored.

In summary, it remains unclear whether unimodal duration tuning changes across timing areas in a way that serves distinct functional roles, follows specific dependencies across areas, or directly contributes to perception. As a result, evidence has yet to converge into a comprehensive framework that accounts for the full transformation of millisecond durations from a stimulus feature (i.e., input) into a flexible perceptual object for decision-making and behavior (i.e., output).

To address these issues, in this study, we asked 13 healthy human participants to perform a visual duration categorization task while we measured brain activity using ultra-high-field (7T) fMRI. We then combined the classical fMRI localization approach with neuronal-based modeling to identify and characterize duration-tuned populations across a multitude of timing areas defined with high anatomical precision. Specifically, we aimed to determine (i) how the properties of unimodal tuning change across the cortical hierarchy; (ii) whether these changes are related across different areas, and (iii) whether they are linked to duration perception. Ultimately, our goal was to define a functional cortical hierarchy for temporal processing and perception in the visual modality.

## Results

To conduct this study, we used fMRI data already available within our research group [11], which met our requirements in terms of experimental manipulation and MRI acquisition (i.e., spatial and temporal resolution, brain coverage). In the experiment, participants were presented with a visual stimulus (i.e., a circular colored Gaussian noise patch subtending 1.5° of visual angle) whose display duration varied pseudo-randomly between 0.2 and 0.8 s. Participants were asked to judge whether the presented duration (i.e., the comparison duration) was longer or shorter than a previously internalized reference duration of 0.5 s. Stimuli were displayed in the lower half of the visual field, either 0.9° or 2.5° of visual angle to the left or right of a central fixation cross, and their spatial location was task irrelevant. Fig 1A shows a schematic representation of the trial structure with a highlight of the 4 spatial positions. Each participant completed 10 blocks (48 trials per block, 2 trials for each combination of stimulus duration and position) acquired in separate fMRI runs. See *Methods - Stimuli and procedure* for additional information.

We conducted a first-level general linear model (GLM) analysis (see *Methods - General linear model (GLM) analysis*), modeling the offsets of the 24 unique combinations of comparison durations and positions as events of interest. The resulting GLM beta weights were then used to feed a vertex-wise modeling procedure based on the population receptive field (pRF) approach [28]. To specifically target neuronal populations selective for temporal information regardless of spatial information, we designed a model that assumes unimodal tuning for stimulus duration, and is invariant to stimulus position (see *Methods - Population receptive field (pRF) modeling*). The model describes the BOLD response of each vertex with two parameters: $\mu_d$, the duration evoking the greatest neuronal response (i.e., duration preference), and $\sigma_d$, the sensitivity of the response. Fig 1B illustrates an example of the model's response function, alongside the corresponding pRF modeling result. All the analyses presented in this work focused on the duration preference ($\mu_d$) parameter.

We restricted our analyses to a set of cortical regions of interest (ROIs) that, according to a group-level GLM analysis, showed a significant activation at the offset of the 6 presented durations (activations were $p < 0.001$ FWE-corrected for multiple comparisons at cluster-level). Locations were identified employing two highly-parcellated atlases (see *Methods - Regions of interest (ROIs) identification*). This approach enabled us to explore all cortical locations recruited for duration processing in our experimental paradigm with high anatomical precision. To provide a comprehensive picture, we also grouped the ROIs into 9 functional streams and conducted all subsequent analyses at both stream and ROI levels. We identified 6 areas in the ventral visual (VV) stream (bilateral V4, left V8, bilateral PIT, bilateral FFC, bilateral VVC, bilateral TE2p); 9 areas in the lateral visual (LV) stream (bilateral V3CD, right LO1, bilateral LO2, right LO3, right V4t, bilateral MT, bilateral MST, bilateral FST, bilateral PH); 11 areas within or adjacent to the IPS (bilateral V3B, left V7, bilateral IP0, left IPS1, left 7PL, left VIP, bilateral MIP, bilateral LIPv, bilateral LIPd, left IP2, bilateral AIP); 8 areas in the inferior parietal (IP) lobule (right PGp, right TPOJ3, bilateral TPOJ2, bilateral TPOJ1, right STV, left PFt, left PFop, left OP1); 5 motor and somatosensory (mot-som) areas, all in the left hemisphere (area 1 - hand subdivision, area 3b - hand subdivision, area 3a -

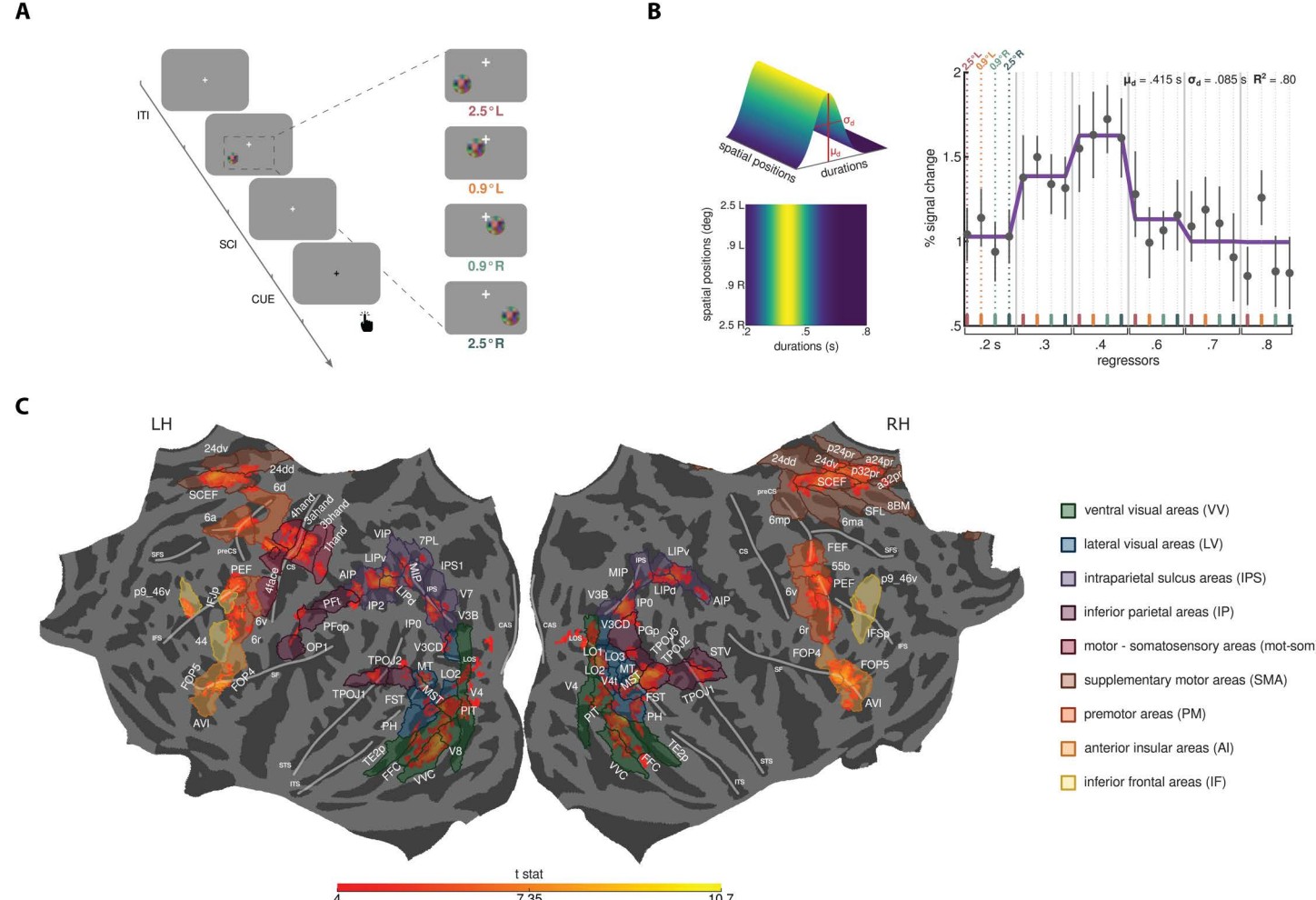

**Fig 1. Experimental procedure, pRF modeling, and regions of interest (ROIs). (A)** Schematic representation of the trial structure. In each trial, one of 6 different comparison durations (i.e., 0.2, 0.3, 0.4, 0.6, 0.7, 0.8 s) was presented at a specific location on the screen, which could be either 2.5° or 0.9° of visual angle in the lower-left (L) or lower-right (R) quadrant of the visual field. Durations varied pseudo-randomly across trials, whereas positions varied sequentially, from 2.5° L to 2.5° R and back. Participants compared the comparison duration to a reference duration of 0.5, which they internalized during the training, and reported by a button press whether the comparison was longer or shorter than the reference. After a randomized interval from the offset of the comparison stimulus (stimulus-cue interval - SCI, uniformly drawn between 0.9-1.2 s), the response was cued with a color switch in the fixation cross from white to black. Trials were interleaved by a uniformly distributed inter-trial interval (ITI) spanning from 1.8 to 2.5 s. The fixation cross was displayed at the center of the screen throughout the experiment. See *Methods - Stimuli and procedure*. **(B)** Result of the pRF modeling in one representative vertex. On the left, three-dimensional (top) and two-dimensional (bottom) representations of the pRF model are shown. Model parameters, i.e., its preference ($\mu_d$) and its sensitivity ($\sigma_d$), are highlighted in the three-dimensional representation. On the right, the plot shows the model fit. The solid line indicates the model prediction. Black dots represent the median GLM beta weights in percent signal change for each combination of stimulus duration and spatial position (color-coded), with error bars indicating standard errors. The top right corner of the plot reports the parameters and goodness of fit of the model prediction. See *Methods - Population receptive field (pRF) modeling*. Source data are available at the following link: osf.io/2tequ [25]. **(C)** Regions of interest (ROIs) are displayed on a common (fsaverage) flattened cortical surface, overlaid on group-level *t*-value clusters ($p < 0.001$ FWE-corrected for multiple comparisons at the cluster level) obtained from a GLM analysis that identified cortical locations significantly activated at the offset of the 6 presented durations. *T*-statistics values are color-coded from red (*t*-stats = 4) to yellow (*t*-stats = 10.7). The HCP MMP 1.0 atlas [26] and the topological atlas by Sereno and colleagues [27] were used to localize the *t*-value clusters, resulting in 47 ROIs in the left hemisphere and 46 ROIs in the right, with 29 ROIs shared between hemispheres. See *Methods - Regions of interest (ROIs) identification* and S1 Fig. ROIs are labeled in white and follow the nomenclature of their respective atlases. They are color-coded according to functional streams: green for ventral visual (VV), blue for lateral visual (LV), violet for IPS, purple for inferior parietal (IP), red for motor-somatosensory (mot-som), brown for SMA, orange for premotor (PM), ochre for anterior insula (AI), yellow for inferior frontal (IF). Major sulci are displayed as thick semi-transparent white lines, with the following abbreviations: CAS = calcarine sulcus; LOS = lateral occipital sulcus; ITS = inferior temporal sulcus; STS = superior temporal sulcus; IPS = intraparietal sulcus; SF = Sylvian fissure; CS = central sulcus; IFS = inferior frontal sulcus; SFS = superior frontal sulcus; preCS = precentral sulcus.

hand subdivision, area 4 - hand subdivision, area 4 - face subdivision); 11 areas in the SMA (right 6mp, right 6ma, bilateral 24dd, bilateral 24dv, right p24pr, right SFL, bilateral SCEF, right 8BM, right p32pr, right a32pr, right a24pr); 7 premotor (PM) areas (left 6d, left 6a, right FEF, right 55b, bilateral PEF, bilateral 6v, bilateral 6r); 3 areas in the anterior insular (AI) region (bilateral FOP4, bilateral FOP5, bilateral AVI); 4 areas of the inferior frontal (IF) lobule (left IFJp, left 44, right IFSp, bilateral p9_46v). Fig 1C shows all the ROIs on a common surface, color-coded according to the functional stream they belong to.

**How duration preferences change along the cortical hierarchy**

To obtain an initial indication of changes in duration tuning along the cortical hierarchy, we assessed how preferred durations vary across functional streams and ROIs (see *Methods - Analysis of duration preference changes along the cortical hierarchy*). Fig 2 shows preferred duration maps for a selection of ROIs across all participants, while Fig 3 illustrates the vertex-wise distributions at group level of preferred durations for each functional stream (A) and ROI (B).

To quantify changes in duration preferences across functional streams, we tested a linear mixed effect (LME) model with stream as factor, and ROI and participant as random intercepts, using individual median $\mu_d$ for each ROI (model formula: $\mu_d \sim stream + (1|ROI) + (1|subjectID)$, marginal $R^2$: 0.09, conditional $R^2$: 0.16). Type III ANOVA on model estimates revealed a main effect of stream ($F(8,55) = 6.73$ $p < 0.0001$). Specifically, the median preferred duration was higher in occipital visual streams compared to parietal and frontal ones (all $t(55) > 4.62$, $p < 0.001$ comparing LV to IPS, IP, mot-som, SMA, and IF; all $t(55) > 3.44$, $p < 0.05$ comparing VV to mot-som and IF). Similarly, to quantify changes in duration preferences across ROIs, we tested a LME model with ROI as factor and participant as random intercept, using individual median $\mu_d$ for each ROI (model formula: $\mu_d \sim ROI + (1|subjectID)$, marginal $R^2$: 0.18, conditional $R^2$: 0.20). Type III ANOVA on model estimates revealed a main effect of ROI ($F(63,756) = 2.98$ $p < 0.0001$). In LO1 and LO2 (belonging to the LV stream) the median $\mu_d$ was significantly higher than in several ROIs within parietal, supplementary motor, and inferior frontal regions (all $t(756) > 4.55$, $p < 0.05$ comparing LO1 to 7PL, VIP, OP1; all $t(756) > 4.32$, $p < 0.05$ comparing LO2 to IPS1, 7PL, VIP, OP1, 8BM, 24dd, 24dv, IFJp, IFSp, p9_46v). In motor and somatosensory hand areas, the median $\mu_d$ was significantly lower than in many occipital visual areas (all $t(756) > 4.31$, $p < 0.05$ comparing areas 1_hand, 3a_hand, and 4_hand to V4, PIT, V3CD, LO1, LO2, V4t, and V3B). The full set of statistics is reported in S1–S4 Tables. All reported $p$-values were Bonferroni-corrected for multiple comparisons.

In line with the statistical assessments, the $\mu_d$ distributions shown in Fig 3 are mainly skewed towards longer preferences in the ventral and lateral visual streams (VV and LV), whereas they shift towards shorter preferences in motor and somatosensory cortices (mot-som). Interestingly, in the IPS and inferior parietal (IP) areas, the distributions are more evenly spread across the duration range, while from SMA onward they are mainly centered around the mean of the duration range. At the ROI level, this pattern remains evident (see also Fig 2), although not all ROIs within each stream show consistency. The highest variability among ROIs is observed in the IPS, IP, and SMA streams. These findings highlight significant changes in the distribution of duration preferences across areas, suggesting that response properties of duration-tuned populations may vary throughout the cortical hierarchy.

**Categorization of duration preferences along the cortical hierarchy**

The previous analysis of $\mu_d$ distributions suggested that a key difference between cortical areas may be the temporal range they are tuned to (i.e., either the full range of presented durations or a narrower range around a specific duration). These different ranges of duration selectivity may thus indicate distinct processing stages of stimulus temporal information. To address this, we grouped duration preferences into 5 categories, each corresponding to a different portion of the tested range: short (0.2−0.32 s), mid-short (0.32−0.44 s), medium (0.44−0.56 s), mid-long (0.56−0.68 s), and long (0.68−0.8 s) (see *Methods - Analysis of duration preference categories along the cortical hierarchy - Categorization of duration preferences*). For each participant, we then computed the fraction of vertices within each duration category for

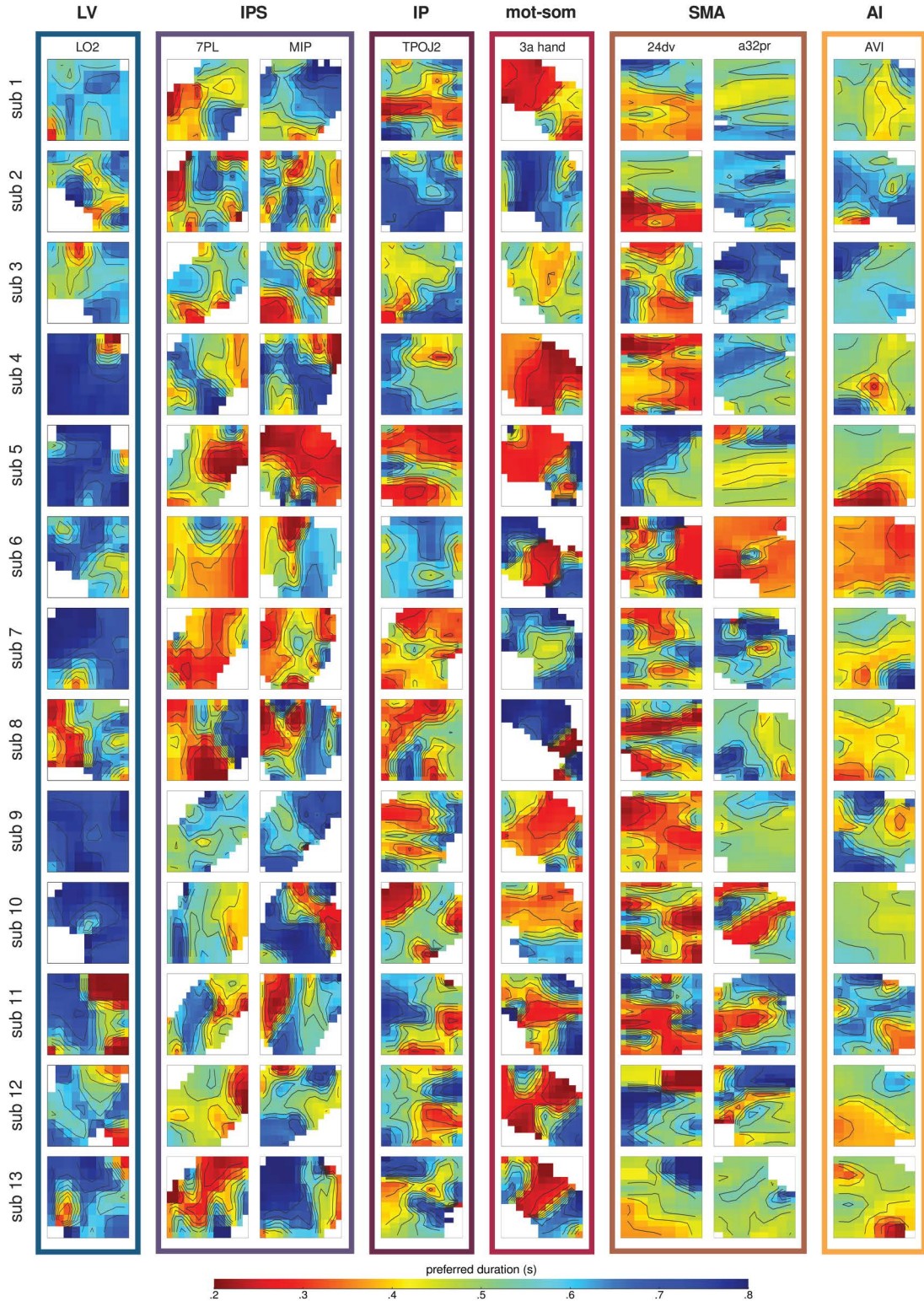

**Fig 2. Maps of duration preferences.** Duration preference maps are shown for all participants (in separate rows) across a selection of ROIs (ordered from occipital to frontal in the different columns), in either the left (LO2, 7PL, MIP, TPOJ2, 3a_hand, AVI) or the right (24dv, a32pr) hemisphere. Each map represents the cortical space of a given ROI, flattened and resampled onto an isotropic two-dimensional grid (2 mm resolution). The color of

each grid cell represents the average duration preference (from red, 0.2 s, to blue, 0.8 s) of the ROI vertices weighted by a Gaussian kernel (full-width half-maximum = 4 mm) centered on that cell. Maps belonging to the same functional stream are grouped with a color-coded outline (blue for LV, violet for IPS, purple for IP, red for mot-som, brown for SMA, ochre for AI). Black lines indicate isolines that join locations with equal preference values. White pixels correspond to locations outside of the ROI. Source data are available at the following link: osf.io/2tequ [25].

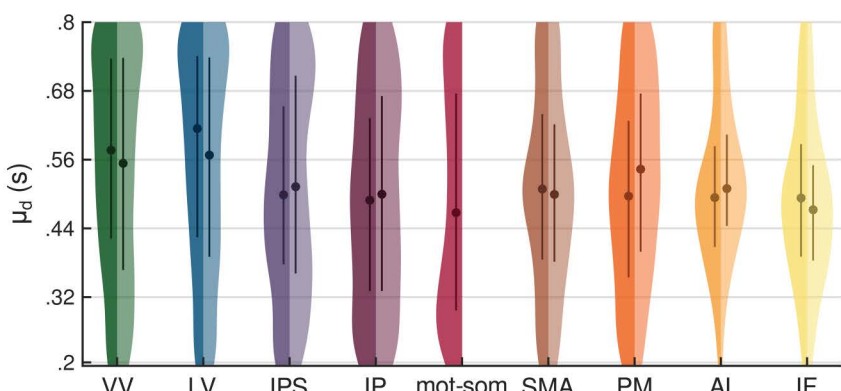

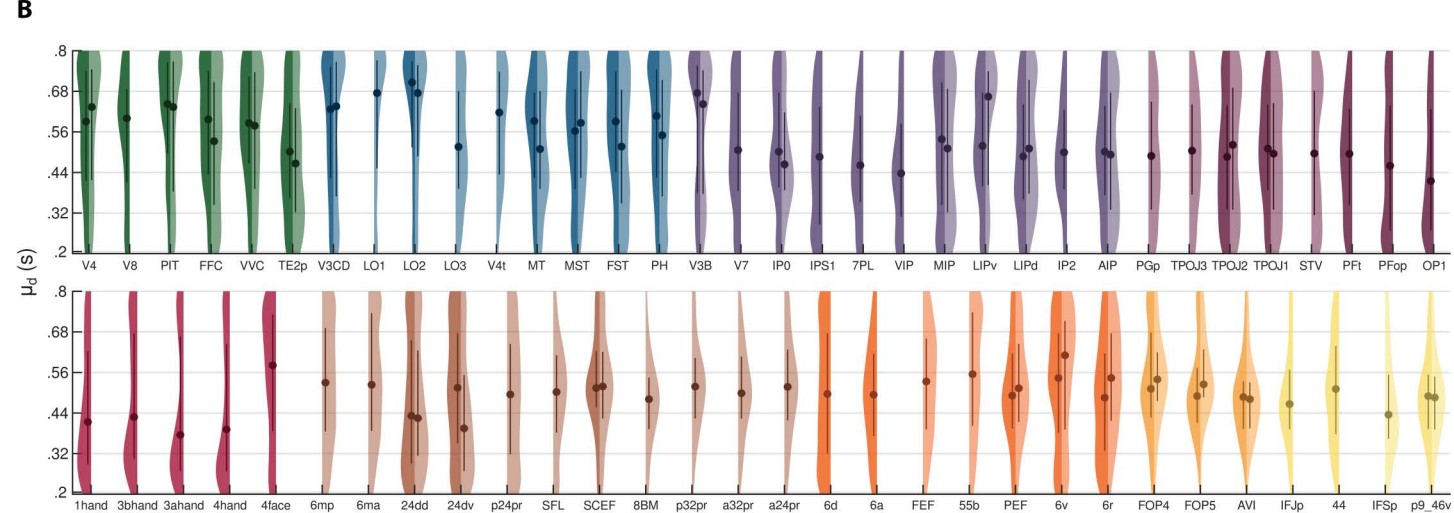

**Fig 3. Group-level distributions of duration preferences.** Each violin plot shows the vertex-wise distribution of duration preferences at group level across streams **(A)** and ROIs **(B)**. Streams and ROIs are ordered from occipital to frontal and from dorsal to ventral. Violins are color-coded according to functional streams: green for VV, blue for LV, violet for IPS, purple for IP, red for mot-som, brown for SMA, orange for PM, ochre for AI, yellow for IF. The left side of the violins (darker shades) represents the left hemisphere, while the right side (lighter shades) represents the right hemisphere. Dots indicate the median of each distribution, thick lines represent the interquartile range. The kernel density estimate of each distribution was computed using a 10% bandwidth. See *Methods - Analysis of duration preference changes along the cortical hierarchy*. Source data are available at the following link: osf. io/2tequ [25].

each functional stream and ROI, averaging across hemispheres for bilateral ones. Fig 4 shows the group-level fractions of vertices across streams (A) and ROIs (C). These data were analyzed using two separate two-way repeated-measures ANOVA, with category and either stream or ROI as within-subject factors. The ANOVA showed a significant main effect of category (across streams: $F(4,48) = 5.45$, $p < 0.005$; across ROIs: $F(4,48) = 4.38$, $p < 0.005$) and a significant interaction

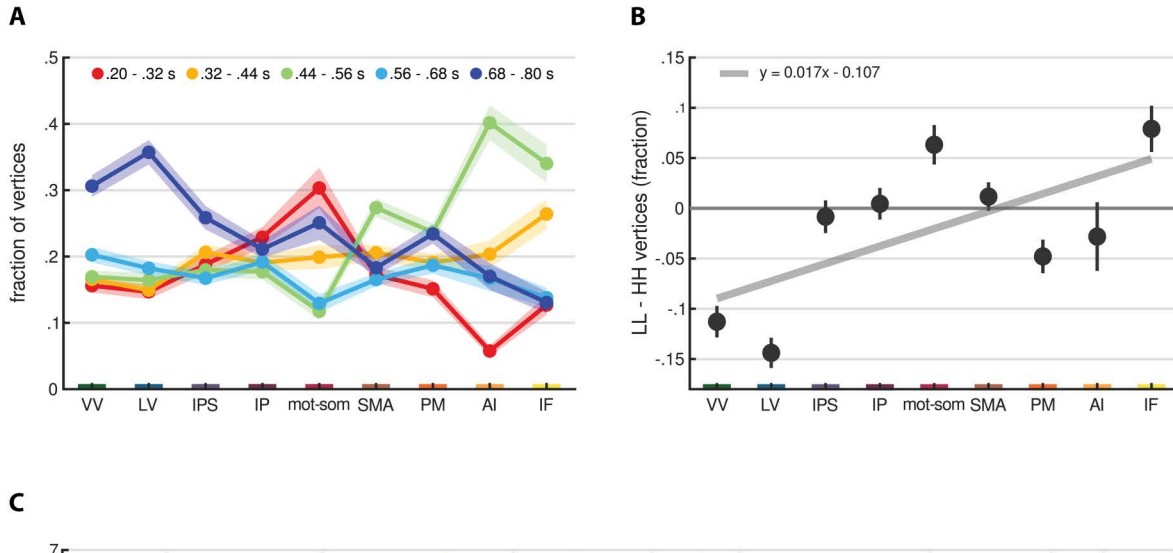

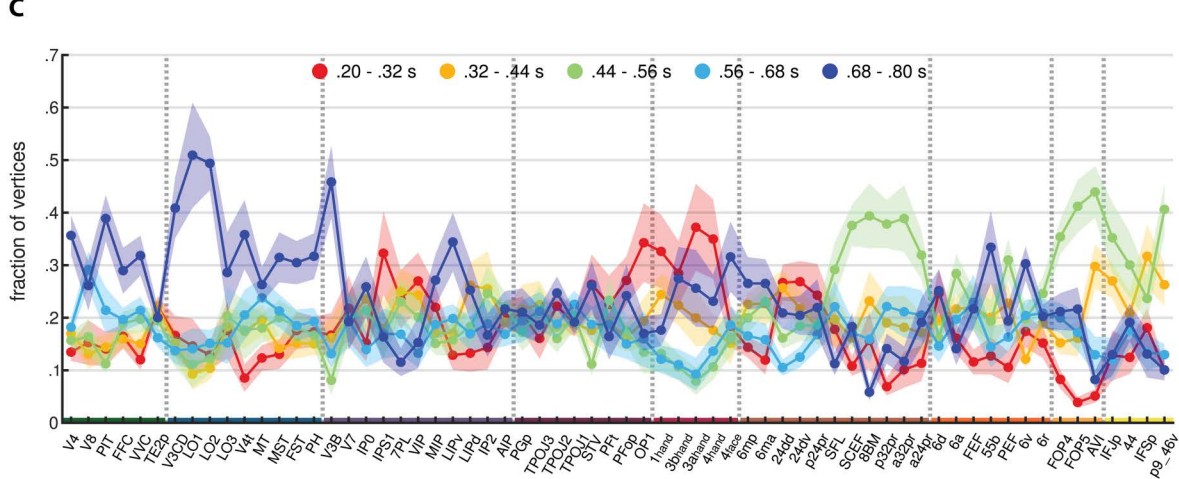

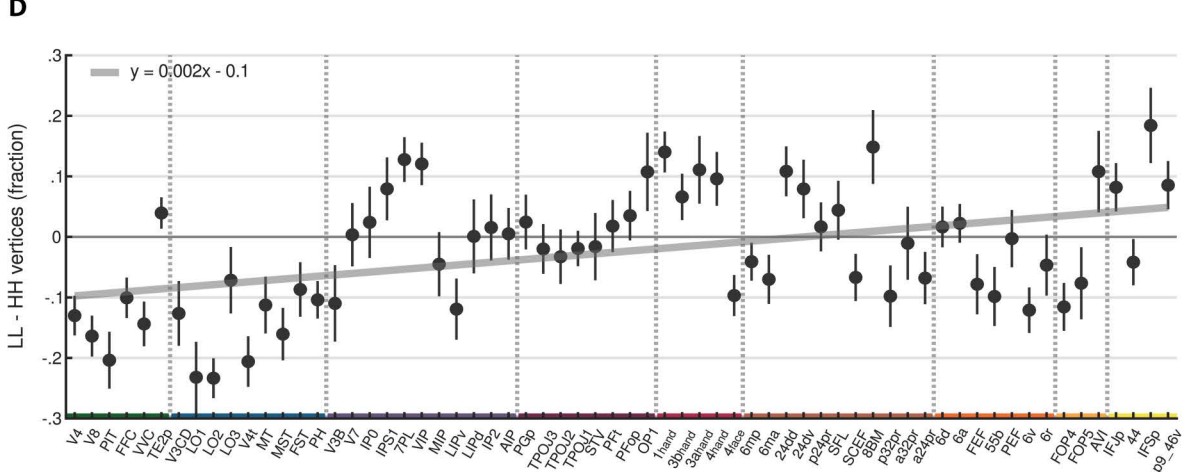

**Fig 4. Categorization and local Moran's *I* of duration preferences.** Panels **(A)** and **(C)** show the group-level fractions of vertices (y-axis) assigned to each duration category across streams and ROIs (x-axis), respectively. Each color represents a different duration category: red for short (0.2−0.32 s), yellow for mid-short (0.32−0.44 s), green for medium (0.44−0.56 s), light-blue for mid-long (0.56−0.68 s), blue for long (0.68−0.8 s). Dots indicate

the mean fraction across participants and hemispheres and are connected by lines for visualization purposes. Shaded areas represent standard errors. Panels **(B)** and **(D)** show the group-level differences (y-axis) in the fraction of vertices between low-low (LL) and high-high (HH) spatial associations across streams and ROIs (x-axis), respectively. These spatial associations were computed using the local Moran's *I* statistic. Dots represent the mean difference across participants and hemispheres, with error bars indicating standard errors. Regression lines are displayed in gray, and their equations are shown in the top left corner of the corresponding plot. Across all panels, streams and ROIs are ordered from occipital to frontal and from dorsal to ventral. Streams are color-coded on the x-axis: green for VV, blue for LV, violet for IPS, purple for IP, red for mot-som, brown for SMA, orange for PM, ochre for AI, yellow for IF. Vertical dashed lines in panels **(C)** and **(D)** separate different streams. See *Methods - Analysis of duration preference categories along the cortical hierarchy*. Source data are available at the following link: osf.io/2tequ [25].

between category and either stream or ROI (across streams: $F(32,384) = 6.40$ $p < 0.0001$; across ROIs: $F(252,3024) = 3.15$ $p < 0.0001$), indicating that duration categories are represented differently across brain regions. As also shown in Fig 4A, in the ventral and lateral visual streams (VV and LV), the long category was the most represented among all others ($t(540) > 3.39$, $p < 0.01$ in VV comparing long with short, mid-short, and medium; all $t(540) > 4.06$, $p < 0.001$ in LV), while in the anterior insula (AI) and inferior frontal cortex (IF), the medium category prevailed (all $t(540) > 4.60$, $p < 0.0001$ in AI; $t(540) > 4.93$, $p < 0.0001$ in IF comparing medium with short, mid-long, and long). In the motor and somatosensory areas (mot-som) instead, the short category prevailed over the medium and the mid-long ones ($t(540) > 4.24$, $p < 0.0005$). Interestingly, in the other streams, all duration categories were equally represented. A similar pattern of results was observed at the ROI level, although the finer resolution allowed us to identify some exceptions. In V3B—the most occipital ROI of the IPS stream, located at its posterior limit—the long category was the most represented, similar to the ROIs in the ventral and lateral visual streams (all $t(3840) > 4.20$, $p < 0.0005$). Likewise, in OP1, a region of the parietal operculum likely contributing to somatosensory processing [29,30], short durations were prevalent, resembling motor and somatosensory areas ($t(3840) > 2.94$, $p < 0.05$ comparing short with medium and mid-long). Another interesting observation concerns the ROIs within SMA, which appeared to separate into two subgroups. Rostral ROIs mainly showed preferences for the medium range (SCEF: $t(3840) > 3.02$, $p < 0.05$ comparing medium with short, mid-short, and mid-long; SFL: $t(3840) = 2.93$, $p < 0.05$ comparing medium with long; 8BM: $t(3840) > 3.62$, $p < 0.005$ comparing medium with short, mid-long, and long; p32pr: $t(3840) > 3.17$, $p < 0.05$ comparing medium with short, mid-short, and long; a32pr: $t(3840) > 3.33$, $p < 0.01$ comparing medium with short, mid-short, and long; a24pr: $t(3840) = 3.80$, $p < 0.005$ comparing medium with short), while caudal ROIs showed no significant differences across duration categories. The complete set of statistics is reported in S5–S12 Tables. All reported *p*-values were Bonferroni-corrected for multiple comparisons. These findings indicate that, along the cortical hierarchy, different areas are selective to different duration ranges. In occipital visual areas, preferences for long durations likely reflect a monotonic tuning for stimulus duration, as already reported in previous studies [11,15,31], which may mediate the encoding of temporal information. In parietal, premotor, and caudal SMA regions, preferences across the full range of presented durations suggest instead the presence of duration-tuned populations that read out the temporal information at hand [11,12]. Finally, inferior frontal regions, anterior insula, and the rostral subdivisions of SMA, by representing the mean of the duration range, may provide the categorical boundary employed for the categorization task [32]. In motor and somatosensory areas, the selectivity for short durations is likely a byproduct of motor preparation responses, as supported by the observed negative relationship between brain activity preceding the response and reaction times (see S13–S15 Tables and *Discussion*). Collectively, these findings point to three distinct stages of temporal processing, each characterized by different properties of duration tuning and implemented in specific cortical areas: duration encoding, duration readout, and duration categorization.

To further validate the previous analysis, we explored the spatial association between neighboring duration preferences by computing the local Moran's *I* statistic. This statistic is computed, for each vertex in the brain, as the correlation between its duration preference and the average preference of its neighbors. In this way, it identifies vertices whose duration preference is significantly associated with their neighbors and classifies the resulting association into the following types: high duration preference surrounded by high ones (high-high, or HH); low surrounded by low (low-low,

or LL); high surrounded by low (high-low, or HL); and low surrounded by high (low-high, or LH). Before computing the Moran's $I$, we centered duration preference values on the mean preference of the hemisphere. Therefore, the terms "low" and "high" are relative to this average, regardless of the absolute deviation. For further details, see *Methods - Analysis of duration preference categories along the cortical hierarchy - Local Moran's I*. We hypothesized that brain regions selective to one specific duration category (i.e., VV, LV, mot-som, rostral SMA, AI, IF) would also show one dominant type of spatial association between neighboring vertices. On the other hand, regions where all duration categories are equally represented (i.e., IPS, IP, caudal SMA) should show multiple spatial association types. To test this, for each participant we computed the fraction of vertices showing each association type within each ROI. Interestingly, the HL and LH associations were nearly absent. On average across participants, hemispheres, and ROIs, their fractions were less than 0.019 (see also S2 Fig), indicating that similar duration preferences tend to cluster together. For this reason, we focused the following analyses on the LL and HH associations only, comparing their respective fraction of vertices. Fig 4 shows the group-level differences across streams (B) and ROIs (D). To assess changes in the LL-HH difference across functional streams, we used a LME model with stream as factor, and ROI and participant as random intercepts (model formula: $LL - HH \sim stream + (1|ROI) + (1|subjectID)$), marginal $R^2$: 0.09, conditional $R^2$: 0.15). Similarly, to quantify changes across ROIs, we used a LME model with ROI as factor and participant as random intercept (model formula: $LL - HH \sim ROI + (1|subjectID)$), marginal $R^2$: 0.17, conditional $R^2$: 0.19). Type III ANOVA on model estimates revealed a main effect of both stream ($F(8,55) = 6.47$, $p < 0.0001$) and ROI ($F(63,756) = 2.81$, $p < 0.0001$), indicating a systematic change in the LL-HH difference along the cortical hierarchy (see S16 and S17 Tables). Specifically, this difference increased along the cortical hierarchy, shifting from negative to positive values ($\beta = 0.017$, $t(7) = 2.25$, $p = 0.059$ across streams; $\beta = 0.002$, $t(62) = 3.79$, $p < 0.0005$ across ROIs). In occipital regions (nearly all ROIs within VV and LV), HH associations were more frequent than LL, whereas in inferior frontal and anterior insular regions (AVI, IFJp, IFSp, p9_46v) the opposite happened. ROIs in the motor-somatosensory stream also showed a prevalence of LL associations. Interestingly, several ROIs within the IPS, inferior parietal lobule (IP), and SMA showed a LL-HH difference close to 0, indicating an equal presence of both spatial associations. See S18–S23 Tables for additional supporting statistics. These results are in line with the previous findings from the categorization analysis of duration preferences. Areas selective for a specific duration category also showed one main type of spatial association between neighboring vertices: HH in occipital visual areas, where long preferences predominated, and LL in motor-somatosensory regions and inferior frontal cortex, where preferences were mainly for low and medium durations, respectively. In regions showing the full range of duration preferences, such as parietal and supplementary motor regions, HH and LL associations coexisted. Overall, these results suggest that visual duration processing is supported by both duration selectivity and the local cortical arrangement of duration information. Importantly, different duration tuning properties emerge at different levels of the cortical hierarchy. These findings strengthen the idea that duration tuning contributes to the transformation and representation of duration information across the cortical hierarchy by mediating different stages of temporal processing.

## The topographic organization of duration preferences along the cortical hierarchy

A key property of unimodal responses to duration is their organization into topographic maps along the cortical surface [12–14]. Our next goal was to investigate whether and how the topographic properties of duration preferences vary across areas, gaining further details about the changes of unimodal tuning along the cortical hierarchy.

First, we tested whether duration preferences were spatially clustered within each ROI by computing the global Moran's $I$ statistic (see *Methods - Analysis of the topographic organization of duration preferences along the cortical hierarchy - Global Moran's I*). The global Moran's $I$ quantifies the overall spatial autocorrelation of duration preferences within the ROI and is computed by averaging all local Moran's $I$ statistics (i.e., the correlation between each vertex's duration preference and the average preference of its neighbors). A global Moran's $I$ close to 0 indicates a random distribution of duration preferences along the cortical surface. Values above or below zero indicate different degrees of spatial clustering,

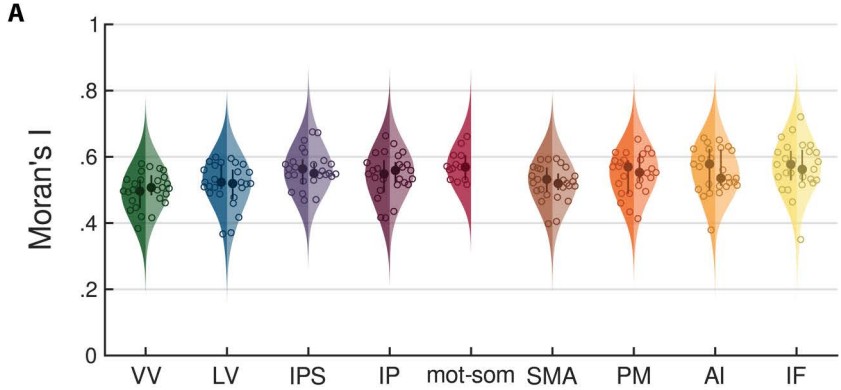

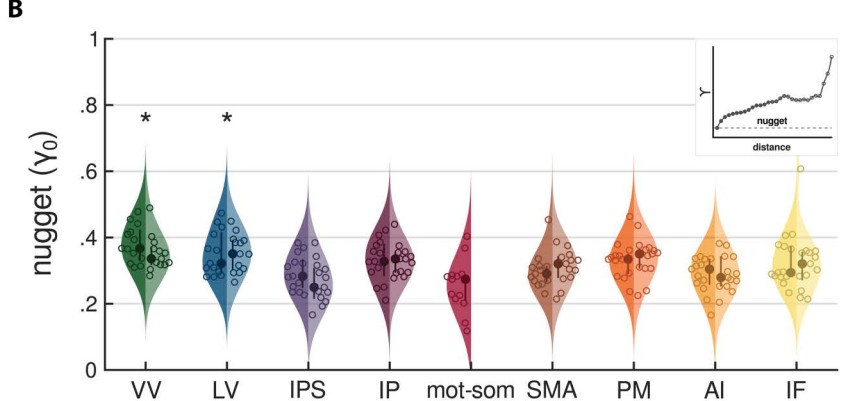

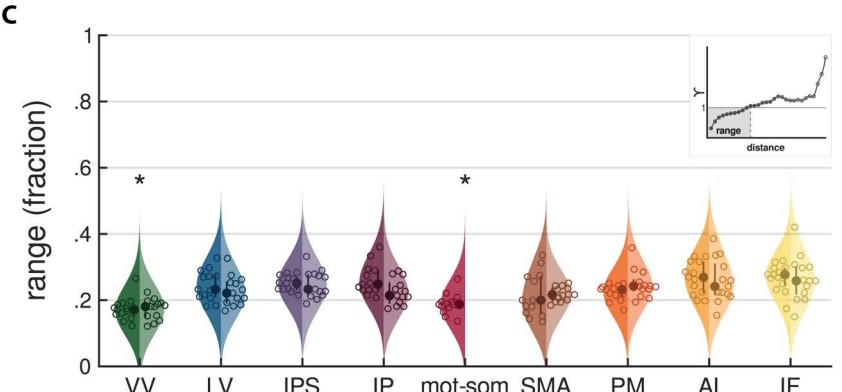

**Fig 5. Group-level distributions of Moran's *I*, nugget, and range values.** Each violin plot represents the group-level distribution of Moran's *I* statistic **(A)**, and variogram nugget **(B)** and range **(C)** values across streams. Nugget values are expressed as a fraction of the total variance within the ROI; range values are expressed as a fraction of the maximum distance between vertices of the ROI. Streams are ordered from occipital to frontal and from dorsal to ventral, and are color-coded as follows: green for VV, blue for LV, violet for IPS, purple for IP, red for mot-som, brown for SMA, orange for PM, ochre for AI, yellow for IF. The left side of each violin represents the left hemisphere (darker shades), while the right side represents the right hemisphere (lighter shades). Dots indicate the median of each distribution, while circles correspond to individual data points. Thick lines represent interquartile ranges. The kernel density estimates were computed using a 7% bandwidth. In panels B and C, asterisks indicate streams that differ statistically from the others. An example variogram highlighting the nugget and the range is shown as an inset in panels **(B)** and **(C)**. In the inset graph, each dot represents the variance (γ) in duration preferences between pairs of vertices at increasing distance. Dots shading reflects the number of vertex pairs at each distance, with darker shades indicating a higher count. The solid line (γ=1) in C indicates the variance in duration preferences across all vertices, without accounting for spatial structure. The nugget (i.e., the variance at the shortest vertex distance) is marked in B by the dashed line, whereas in C the range (i.e., the distance at which the total variance is reached) is highlighted by a gray box. See *Methods - Analysis of the topographic organization of duration preferences along the cortical hierarchy*. Source data are available at the following link: osf.io/2tequ [25].

characterized by either a positive (i.e., clustering of similar duration preferences) or a negative (i.e., clustering of opposite preferences) spatial autocorrelation. Fig 5A shows the group-level distributions of this statistic across streams, while S3 Fig shows group-level Moran's *I* distributions across both streams (A) and ROIs (B). Due to the unique neighborhood configuration of each ROI, statistical comparison of Moran's *I* values across ROIs is not feasible. The results revealed a positive and relatively stable Moran's *I* across the cortical hierarchy (median across participants above 0.50 in all streams and above 0.34 in all ROIs), indicating that vertices with similar preferences are consistently clustered together. However, occipital visual areas (VV and LV) showed slightly lower values than the other regions.

We next computed the experimental variogram for each ROI and extracted two additional indicators of spatial autocorrelation: the nugget and the range (see *Methods - Analysis of the topographic organization of duration preferences along the cortical hierarchy - Variogram*). The variogram estimates spatial autocorrelation by computing the variance among the duration preferences of vertices at varying distances within the ROI. In a variogram, the nugget is the variance observed at the shortest distance between vertices in the ROI (see the inset in Fig 5B), and the range is the distance between vertices required to reach the overall variance of the ROI (see the inset in Fig 5C). While the previously described global Moran's *I* captures the overall spatial pattern within the ROI, the nugget provides finer resolution by focusing on neighboring vertices and reflects the strength of the spatial autocorrelation. The range, in contrast, quantifies the persistence of spatial autocorrelation along the cortical surface. Fig 5 shows the group-level distributions of nuggets (B) and ranges (C) across streams, while S4 and S5 Figs show the group-level distributions of nuggets and ranges respectively, across both streams (B) and ROIs (C). To assess changes of nugget and range across functional streams, we tested two LME models with stream as factor and participant as random intercept. The LME model formulas were as follows: $nugget \sim stream + (1|subjectID)$ (marginal $R^2$: 0.31, conditional $R^2$: 0.57) and $\sqrt{range} \sim stream + (1|subjectID)$ (marginal $R^2$: 0.51, conditional $R^2$: 0.53). Type III ANOVA on model estimates revealed a main effect of stream for both nugget and range (nugget: $F(8,96) = 10.26$ $p < 0.0001$; range: $F(8,96) = 15.76$ $p < 0.0001$). Nuggets were significantly higher in occipital visual streams compared to parietal and frontal ones (all $t(96) > 3.80$, $p < 0.01$ comparing VV to IPS, mot-som, SMA, and AI; all $t(96) > 4.06$, $p < 0.01$ comparing LV to IPS, mot-som, and AI). Conversely, ranges were significantly lower in both ventral visual (VV) and motor-somatosensory (mot-som) streams compared to the majority of the other streams (all $t(96) > 4.14$, $p < 0.01$ comparing VV to LV, IPS, IP, SMA, PM, AI, and IF; all $t(96) > 4.14$, $p < 0.01$ comparing mot-som to LV, IPS, IP, PM, AI, and IF). The complete set of statistics is reported in S24-S27 Tables. All reported *p*-values were Bonferroni-corrected for multiple comparisons. Similar results were obtained performing LME model analyses at the ROI level (see S28-S31 Tables).

In summary, the assessment of topographic properties of duration preferences with Moran's *I*, nuggets, and ranges reveals that occipital (VV and LV) and motor-somatosensory areas exhibit a weaker spatial arrangement compared to the other areas. This suggests that in these regions the spatial organization of duration preferences is less critical for duration processing.

**Long-range relationships between duration preferences**

After characterizing the changes in duration preferences across different brain regions, we next explored whether and how these changes are related along the cortical hierarchy. To address this at the ROI level, for each participant we computed the median duration preference within each ROI and then calculated the group-level Kendall's τ correlation matrix, shown in Fig 6 (see *Methods - Analysis of long-range relationships between duration preferences*). This analysis allowed us to capture the overall pattern of relationships between duration preferences across ROIs. From a qualitative point of view, the correlation matrix showed mainly positive values within functional streams, with higher consistency in visual (VV, LV), motor-somatosensory (mot-som), and anterior insular (AI) streams. This indicates that duration preferences change similarly in ROIs that are in close anatomical and functional proximity. More interestingly, the correlation matrix allowed us to explore relationships between ROIs from distinct and distant streams, providing insight into the hierarchical organization of duration preference changes. We observed negative correlations between ROIs at the opposite ends of the

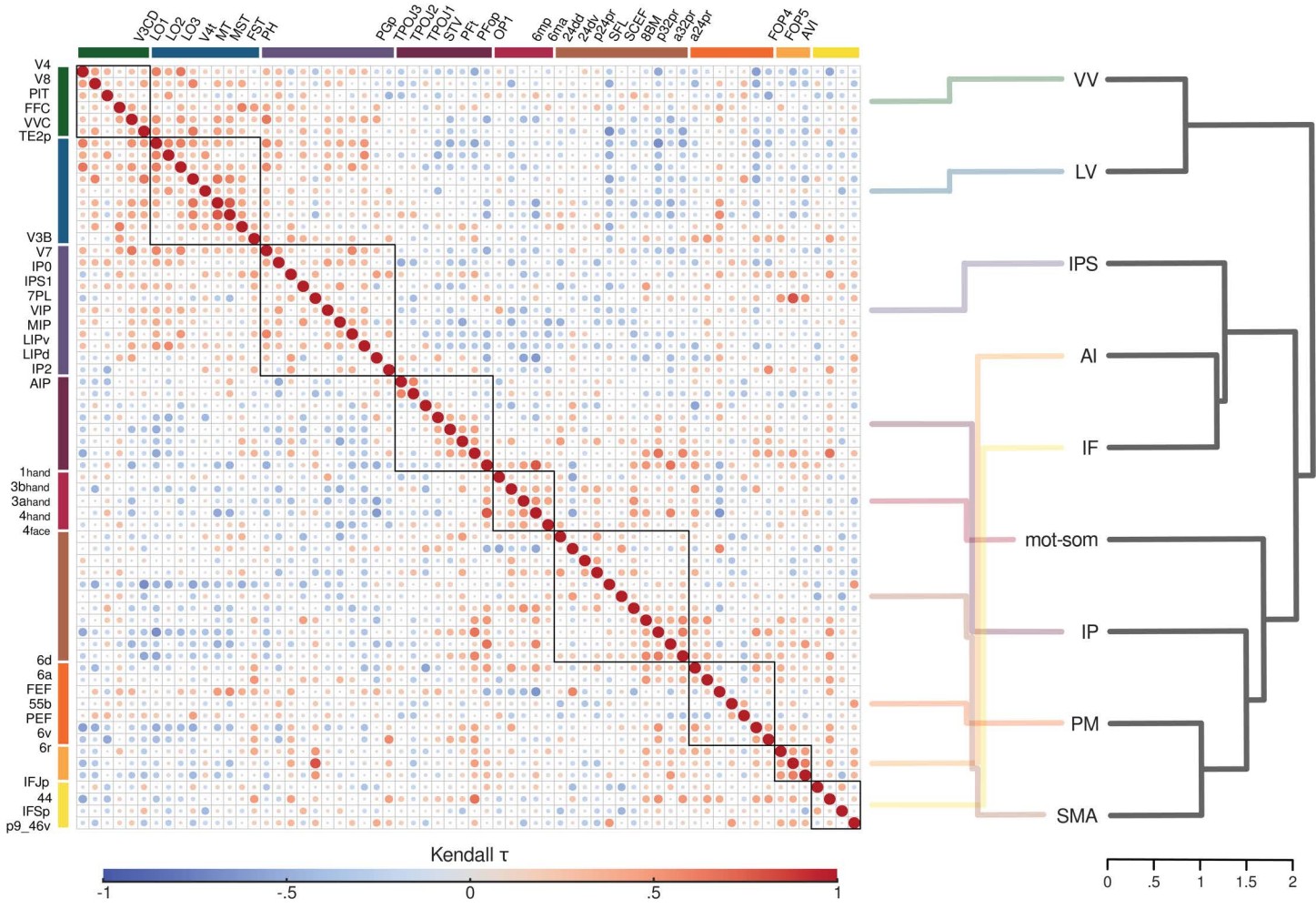

**Fig 6. Long-range relationships between duration preferences.** The left side shows the group-level Kendall's τ correlation matrix, computed using the median preferred duration of each ROI and participant. The size and color of each dot represent the Kendall's τ value. This visualization was generated using the corrplot package [33] in R. ROIs on x and y axes are arranged from occipital to frontal and from dorsal to ventral, with colors indicating their respective streams: green for VV, blue for LV, violet for IPS, purple for IP, red for mot-som, brown for SMA, orange for PM, ochre for AI, yellow for IF. Black squares within the matrix outline ROIs belonging to the same stream. The right side displays the dendrogram representation of the Kendall's τ correlation matrix at the stream level. Semi-transparent lines, color-coded by stream, connect the ROI representation on the left to the stream representation on the right for visualization purposes. See *Methods - Analysis of long-range relationships between duration preferences*. Source data are available at the following link: osf.io/2tequ [25].

hierarchy. Specifically, ROIs in the ventral visual stream (VV) were positively correlated with the nearby ROIs in the lateral visual stream (LV) but negatively correlated with all other ROIs from more distant streams. Similarly, ROIs rostral to the motor-somatosensory stream (mot-som) were positively correlated among themselves but negatively correlated with both occipital streams (VV and LV). For the parietal streams (IPS and IP), correlation patterns were more nuanced. ROIs in the IPS were positively correlated with occipital (VV and LV), anterior insular (AI) and inferior frontal (IF) regions, but mainly negatively correlated with the other streams. By contrast, ROIs in the inferior parietal cortex (IP) appeared to segregate into two groups. PGp and TPOJ3 showed overall weak correlations, but tended to positively correlate with some ROIs in the lateral visual (LV) stream. From TPOJ2, ROIs showed instead negative correlations with caudal ROIs and positive correlations with rostral ROIs.

To better capture and quantify long-range relationships between duration preferences, we recomputed the Kendall's τ correlation matrix using individual median preferred durations of each stream and used these data to build the dendrogram shown in Fig 6 (see *Methods - Analysis of long-range relationships between duration preferences*). The dendrogram represents the similarity between streams based on the correlation of their duration preferences. Streams with stronger positive correlations are placed closer together, while those with weaker or negative correlations are located farther apart, resulting in a hierarchical representation of duration preference correlations. This analysis identified three main clusters. The first cluster included lateral and ventral visual streams (VV and LV). The second cluster included the IPS, the anterior insula (AI), and the inferior frontal cortex (IF). The third cluster comprised the inferior parietal lobule (IP), the medial and lateral premotor cortices (SMA and PM), and the motor-somatosensory cortices (mot-som). Within the second cluster, anterior insula and inferior frontal areas were more similar to each other than to the IPS. Within the third cluster, SMA and premotor cortex showed the strongest similarity, followed by inferior parietal lobule and motor-somatosensory cortex. Interestingly, the closest pairs within each cluster (VV-LV in cluster 1, AI-IF in cluster 2, and SMA-PM in cluster 3), besides being anatomically close, also exhibited similar properties in the previous analyses (i.e., similar categories of preferred durations, similar local spatial associations between duration preferences, and similar degrees of clustering of duration preferences).

Overall, these findings suggest that changes in duration preferences along the cortical hierarchy are shaped by specific, non-trivial relationship between regions. When considered in light of our previous results, they point to a hierarchical organization of duration processing, where different areas showing different tuning properties may work together to support different outcomes of temporal processing. In occipital visual areas (cluster 1), duration is extracted from the sensory input and encoded through the monotonic increase in neuronal response amplitude. The two downstream clusters, instead, include parietal and frontal areas that support both duration readout and categorization. This suggests that these two clusters may contribute in parallel to different aspects of duration perception: one likely related to higher-level cognitive functions, involving the IPS, inferior frontal cortex, and anterior insula (cluster 2); and the other more motor-oriented, engaging the inferior parietal cortex and motor-related regions (cluster 3).

### Linking duration preferences to perception

Finally, we investigated whether and where, along the cortical hierarchy, duration preferences are linked to duration perception in the categorization task. From each participant's psychometric curve (shown in Fig 7A), we derived the Point of Subjective Equality (PSE), defined as the comparison duration that is equally likely to be judged as longer or shorter than the reference duration. The PSE represents, therefore, the subjective boundary each participant employs to perform the task. We then computed the Kendall's τ correlation between individual PSE values ($n$ = 13) and median duration preferences of each individual stream and ROI (see *Methods - Analysis of the link between duration preferences and perception*). Fig 7 shows the correlation across streams (B) and ROIs (C).

At the stream level, PSE values and median duration preferences were significantly and positively correlated in the left inferior parietal cortex (IP, $τ = 0.44$, $p < 0.05$) and in the anterior insula (AI), bilaterally (left: $τ = 0.44$, $p < 0.05$; right: $τ = 0.59$, $p < 0.0001$). When examining individual ROIs, we observed significant positive correlations in the left FOP5, belonging to AI ($τ = 0.43$, $p < 0.05$), in the bilateral AVI, belonging to AI (left: $τ = 0.49$, $p < 0.05$; right: $τ = 0.56$, $p < 0.01$), and in the right a32pr, belonging to SMA ($τ = 0.43$, $p < 0.05$). *P*-values are uncorrected for multiple tests. Interestingly, in these areas the majority of vertices showed, in our previous analyses, a preference for the medium duration range, which already suggested the representation of a categorical boundary used to perform the task. Finding a link between this type of duration preference and PSE further supports this idea and suggests that this representation has a subjective connotation, reflecting each participant's individual perception in the categorization task [32]. The other observed significant correlations are reported in Fig 7C. Finally, to have a comprehensive view of how the correlation between PSE values and preferred durations changed along the cortical hierarchy, we fitted a regression line on z-transformed Kendall's τ values, as shown in

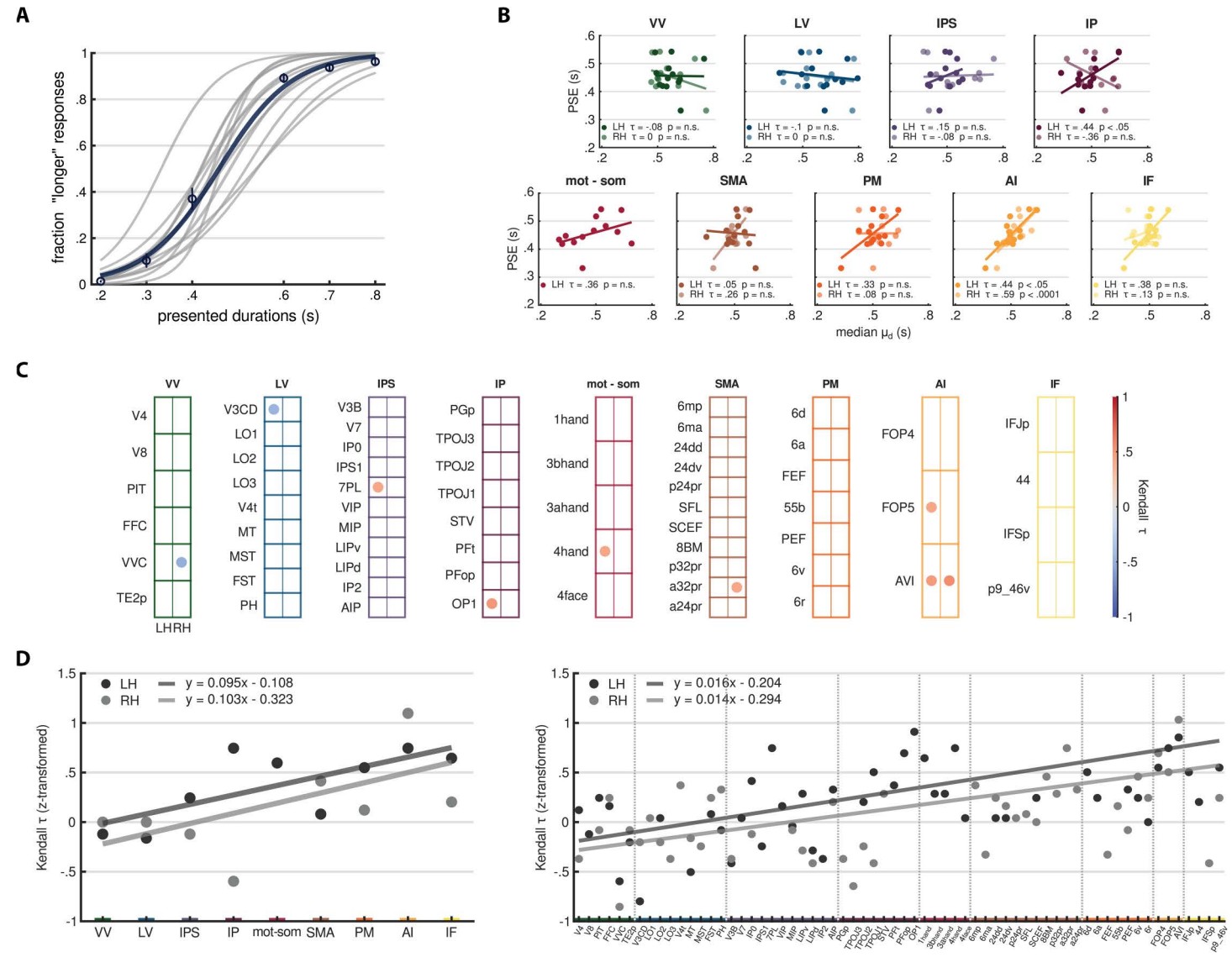

**Fig 7. Correlation between PSE and duration preferences. (A)** The plot shows individual psychometric curves in gray, with the group-level psychometric curve in blue. Circles represent the average fraction of "comparison longer than reference" responses across participants for each comparison duration, with error bars indicating standard errors. **(B)** Each scatter plot corresponds to a different stream, where individual PSE values (y-axis) are plotted against individual median preferred durations (x-axis) of that stream along with regression lines. Darker dots refer to the left hemisphere, while lighter dots refer to the right hemisphere. Kendall's τ correlation values and their associated *p*-values are reported in the bottom left corner of each plot. **(C)** Each grid represents a different stream, with ROIs arranged in rows. The left (LH) and right (RH) hemispheres are displayed in separate columns. Each dot represents the Kendall's τ correlation between individual PSE values and individual median preferred durations for a given ROI. Dot color indicates the Kendall's τ value. Only significant (*p*-value < 0.05) correlations are shown. **(D)** Scatter plots represent z-transformed Kendall's τ values across streams (left plot) and ROIs (right plot). Regression lines are displayed in gray, with their equations shown in the top left corner of each plot. Darker shade represents the left hemisphere, while lighter shade corresponds to the right hemisphere. In the right plot, vertical dashed lines separate different streams. In panels B, C, and D, the color code indicates different streams: green for VV, blue for LV, violet for IPS, purple for IP, red for mot-som, brown for SMA, orange for PM, ochre for AI, yellow for IF. See *Methods - Analysis of the link between duration preferences and perception*. Source data are available at the following link: osf.io/2tequ [25].

Fig 7D, across streams (left plot) and ROIs (right plot). The correlation increased along the cortical hierarchy, shifting from values close to 0 in occipital ROIs to positive values in frontal ROIs (across left streams: $\beta = 0.095$, $t(7) = 2.72$, $p < 0.05$; across right streams: $\beta = 0.103$, $t(6) = 1.49$, $p$-value not significant; across left ROIs: $\beta = 0.016$, $t(45) = 4.59$, $p < 0.0001$; across right ROIs $\beta = 0.014$, $t(44) = 3.59$, $p < 0.001$). This last result suggests that, overall, the link between duration preferences and perception is progressively built up along the cortical hierarchy.

## Discussion

In this study, by means of a neuronal response model of unimodal duration tuning, we investigated how duration-tuned populations change across the cortical hierarchy, whether they are related across different areas, and whether they are linked to duration perception. We focused within cortical locations that showed significant activation at stimuli offset (i.e., when temporal information was fully available to participants), which corresponded to areas commonly reported in neuro-imaging studies using explicit timing tasks of sub-second visual durations [1,3].

Our findings indicate that, along the cortical hierarchy, neuronal populations show unimodal tuning to different ranges of durations. This is evident not only in their specific duration preferences, but also in their local spatial organization. In addition, these duration-tuned neuronal populations display different degrees of global spatial clustering across areas and are directly linked to duration perception only in specific regions. Specifically, our analyses showed that the ventral and lateral visual areas mainly preferred long durations and displayed a lower degree of clustering compared to the other regions. The superior and inferior parietal cortex, lateral premotor cortex, and caudal SMA showed preferences spanning the full range of presented durations, with a higher degree of clustering. The anterior insula, inferior frontal cortex, and rostral SMA mainly preferred durations around the mean of the presented range, also displaying a higher degree of clustering. Notably, within specific areas of this latter group (a32pr within the SMA, and AVI and FOP5 within the anterior insula), duration preferences were positively correlated with PSE values. Finally, duration-tuned neuronal populations appeared to closely co-vary within different subsets of regions. This allowed us to identify three functional clusters: (i) occipital visual areas; (ii) intraparietal, inferior frontal, and insular areas; and (iii) inferior parietal, supplementary motor, premotor, and motor-somatosensory areas. Taken together, these findings suggest that distinct functional stages of duration processing are implemented in specific cortical regions, likely organized hierarchically and supporting different functions.

The first processing stage may occur in ventral and lateral visual areas, where we consistently observed a preference for long durations. This type of preference is compatible with a ramping response to stimulus duration, in line with previous findings showing that neuronal activity in early visual regions increases modestly yet monotonically as a function of stimulus duration [6,11,15,34]. The role of early sensory areas as gateways for time processing and perception has been suggested by diverse evidence, including psychophysics [35–37], brain imaging [11,34–39], and brain stimulation [40,41] works in humans as well as electrophysiology studies in rodents [31,42–44]. In the tactile domain, for example, Reinartz and colleagues [43] showed that optogenetic excitation of the vibrissal somatosensory cortex in rats dilated perceived duration and amplified perceived intensity, suggesting that time perception is deeply rooted in sensory coding. In the visual domain, Tonoyan and colleagues [39] found that temporal biases induced in humans by temporal frequency adaptation could be predicted by the amplitude of an early ERP component (N200) recorded contralaterally to the stimulated visual field. This suggested that the subjective perception of time is linked to processes that begin locally and relatively early in the visual processing stream. In line with this latter study, a recent work from our group also showed that monotonic duration tuning in early visual areas is spatially dependent (i.e., only neuronal populations encoding the stimulus spatial position are modulated by duration) [11]. Collectively, these studies highlight the critical role of early sensory regions in duration processing and perception. Our findings further support this view, suggesting that in the visual domain, monotonic responses may reflect an "accumulation" process of sensory information that encodes the duration of visual events [42]. The negative correlation between duration tuning and behavior observed in specific ventral and lateral and visual areas (right VVC and left V3CD) points to a direct involvement of this accumulation mechanism in duration

perception, although its precise interpretation would require targeted experiments and analyses. Within this framework, such an encoding mechanism could also account for the dilation of perceived time induced by the increase of low-level stimulus properties, such as speed [45], size [46], or visibility [47], by positing that greater sensory evidence accumulation leads to longer encoded durations.

A second functional stage may occur in IPS and inferior parietal areas, lateral premotor regions, and caudal subdivisions of SMA. Here, we observed unimodal responses across the entire range of presented durations, well clustered in topographic maps [12–14]. At this stage, unimodal tuning and its topographic organization may provide the readout of temporal information, that could serve different purposes depending on the location along the cortical hierarchy [48]. On one hand, parietal maps may represent a relay stage, where temporal information from early visual areas is made available for further processing in frontal regions, consistently with the role of parietal cortex in sensory [49] and magnitude [50] binding. In the IPS, for example, different stimulus attributes such as duration, spatial position, size, and numerosity elicit topographically organized unimodal responses [51]. These responses not only coexist but also interact. Indeed, two recent fMRI studies from our group found that neuronal populations in the IPS selectively code for duration and spatial position [11], or duration and numerosity [14]. This suggests that duration readout may serve the integration of temporal information with other sensory information to create a unified stimulus representation. Furthermore, studies using transcranial magnetic stimulation showed the causal involvement of inferior parietal cortex in temporal judgments [41,52], and an indirect measure of unimodal responses to durations in the right supramarginal gyrus has been shown to reflect perceived, rather than physical, durations [21]. These findings collectively suggest that unimodal tuning in parietal areas carries critical information for both sensory and perceptual accounts of temporal processing. On the other hand, temporal maps in lateral premotor areas and caudal SMA may support the readout of temporal information for task implementation and execution. In humans, these regions are widely activated in a variety of timing tasks [1–3]. In particular, the SMA is recognized as a core timing area [53], recruited regardless of duration range, stimulus modality, or task type. SMA activity depends also on the degree of attention directed towards the temporal attribute of the stimulus, with greater attention eliciting greater activity [54]. Indeed, unimodal neuronal responses to durations have been reported in SMA in presence of both perceptual and motor timing tasks, respectively in humans [12] and in monkeys [4,5], but not when participants are only exposed to durations without engaging in judgments or tasks [13,14]. Overall, these findings suggest that unimodal tuning in premotor cortices might be more oriented to behaviorally relevant processing.

A third functional stage may occur in the rostral SMA, anterior insula, and inferior frontal regions, where we observed well-clustered duration preferences around the mean of the presented temporal range. This finding aligns with previous fMRI studies employing passive viewing of changing durations, which showed that, in frontal regions, duration preferences either shift toward the mean of the tested range [14] or become more narrowly distributed [13]. These observations suggest that frontal areas process the overall statistics of the temporal environment and provide an abstract, categorical representation of durations, even when these durations are not immediately required for action. Our results extend this idea by also showing that in specific regions (i.e., a32pr in rostral SMA, FOP5, and AVI in anterior insula) duration preferences positively correlated with individual PSE values. This indicates that duration unimodal tuning at this stage may provide the representation of the perceptual temporal boundary used to solve the task. A similar finding was previously reported by Mendoza and colleagues [32], who found that pre-SMA neurons in monkeys performing an interval categorization task peaked near the boundary between short and long duration categories. A subgroup of these "boundary" neurons also adapted when the categorization criterion changed (i.e., when a different range of durations was tested). Crucially, the activity of these neurons reflected monkeys' PSE values, rather than the physical boundary between categories, suggesting that they encode the subjective decision criterion that guides duration categorization behavior. Our findings suggest the presence of analogous "boundary" neuronal populations in humans, not only within the pre-SMA, but also in the anterior insula, a region associated with awareness, encompassing processes from interoception to perceptual decision-making [55]. Similar to the SMA, the anterior insula is also consistently implicated in timing tasks [3] and has been

proposed to contribute to the subjective experience of duration by integrating interoceptive signals and emotional states over time [17]. Within this framework, Wittmann and colleagues observed that the activity in anterior insula, as well as in the inferior frontal cortex and pre-SMA, increased during the reproduction of supra-second durations, peaking just before the motor response terminating the reproduction [16]. This accumulator-like activity was suggested to reflect a temporal representation for an upcoming decision (i.e., stopping the reproduction), in line with the notion that the accumulation of bodily states and emotions underlies the perception of time [18]. While our experimental approach does not address the embodied aspects of temporal processing, our findings are consistent with the idea that frontal regions contribute to an internal and subjective representation of time. Importantly, we observed that this representation emerges from a cascade of transformations in neuronal tuning properties along the cortex. In addition, our findings suggest that, in the anterior insular cortex, boundary populations may constitute the neural substrate for integrating and directly comparing internal (e.g., bodily signals, temporal priors) and external (i.e., sensory) temporal information.

Finally, motor and somatosensory areas consistently showed preferences for short durations. However, all areas were restricted to the left hemisphere, consistent with the fact that participants provided their responses during the task using their right hand. This lateralized activation, combined with the preference for short durations, suggests that neuronal responses in these areas might reflect motor preparation rather than duration tuning. In each trial, the "ready" signal for motor response likely corresponded to the onset of each stimulus presentation, and among the tested durations, the one closer to the onset of the stimulus is the shortest one. In addition, similarly to occipital cortex, motor and somatosensory areas exhibited an overall weaker spatial autocorrelation compared to the other areas. This may further indicate the absence in these areas of unimodal responses for durations. This observation also strengthens the idea that spatial autocorrelation increases when efficient information processing and transmission are required [56], even when selectivity is restricted to a single duration, as we observed in more frontal areas.

An interesting finding concerns the response properties in the SMA, which are segregated along its rostro-caudal axis. Caudal regions showed unimodal responses to the full range of presented durations, while rostral regions showed preferences for the mean of the range. In addition, in one rostral region (a32pr), duration preferences also correlated with PSE values. The SMA is recognized as a heterogeneous structure, both anatomically and functionally, with a primary subdivision between SMA proper and pre-SMA, located caudal and rostral to the anterior commissure, respectively (for a review, see [57]). A functional rostro-caudal gradient has also been described for temporal processing [2,58,59]. For example, perceptual timing tasks are more likely to activate the pre-SMA, while motor timing tasks are associated with SMA proper [60]. Consistently with this view, the segregation we observed between tuning mechanisms may reflect this broader division of SMA, where a more categorical representation of durations (i.e., unimodal responses for the mean of the range) occurs in rostral areas, associated to more perceptual processing compared to caudal areas. Interestingly, the presence of different tuning mechanisms in the SMA has not been reported previously in humans, though it has been observed in monkeys [4,32]. Overall, this suggests that the SMA may integrate two types of signals from different populations: one providing the readout of the duration of the stimulus at hand, and another representing the (subjective) criterion for solving a timing task. This integration could be a unique property of the SMA that makes it a core site for timing [61].

The study of long-range relationships between duration preferences revealed specific dependencies across brain areas, suggesting a hierarchical organization of the three functional stages described above—encoding, readout, and categorization. We identified three main functional modules where duration-tuned populations appeared to be more closely related: (i) occipital visual areas, (ii) intraparietal, inferior frontal, and insular areas, and (iii) inferior parietal, supplementary motor, premotor, and motor-somatosensory areas. The combination of their duration tuning properties with their involvement in other brain functions suggests that these modules may play different roles in temporal processing. The encoding stage involves occipital visual areas, which likely serve as the entry point for temporal information, where duration is extracted from incoming visual events. In contrast, the readout and categorization stages occur in the two downstream clusters, which engage distinct parietal and frontal areas. In one cluster, duration readout and categorization seem to

function sequentially, with duration-tuned populations in the IPS conveying sensory information to boundary populations in inferior frontal and anterior insular cortices. The IPS, working as a coordinate system for perception [62] with a central role in multisensory [63] and visuomotor processing [64], may integrate temporal information with other sensory inputs [11,14]. This multimodal information may then be used by the inferior frontal and anterior insular cortices to generate a unified and flexible perceptual experience of the sensory environment. This cluster may therefore perform an abstraction of temporal information, potentially through the combination of areas that each implement distinct properties of duration tuning. Consistent with this idea of progressive abstraction, Hayashi and colleagues found that both the intraparietal and inferior frontal cortices store a common representation of temporal and numerical information. However, while the IPS processes this interaction at a perceptual level, the inferior frontal cortex represents it at a more abstract level [65]. The other cluster includes inferior parietal and motor-related regions, with almost all exhibiting neuronal populations that read out temporal information. The SMA is, in fact, the only area where both duration readout and categorization coexist. This cluster may be specialized in the motor implementation of timing decisions and behavior, and the discrete mapping of durations at hand may serve as a framework for immediate actions. The inferior parietal regions in this cluster, beyond their roles in visuospatial processing (PGp) [66] and visuomotor functions (PFop, PFt) [67], also contribute to higher-order cognitive processes such as memory retrieval (PGp) [67], theory of mind (TPOJ1) [68], and emotional regulation (STV) [69] (see also [30,70]). Moreover, areas of the inferior parietal lobule have been causally linked to duration perception [21,41,52,71]. These areas may thus represent temporal information in a behaviorally relevant manner and relay it to downstream motor-related areas, where timing behavior is ultimately implemented, initiated, and controlled.

It is important to note that we limited our investigations exclusively on cortical regions, without considering contributions from the cerebellum and subcortical structures, which are known to play a critical role in temporal processing [72–77]. In addition, we only focused on the visual modality, and the extent to which these findings may generalize to other sensory modalities remains uncertain. For example, in the auditory domain, duration tuning have been found in primary auditory cortices only [78], and a recent review identified different activation patterns for auditory and visual timing [3]. Finally, a proper characterization of "boundary" neuronal populations would benefit from an experimental design with multiple duration ranges [32], which would allow assessing whether their activity shifts, as the PSE does, according to different reference durations in the task. Such a design could also help to better establish the properties and functional role of duration tuning and its topographic organization across the other stages of duration processing.

Overall, this study successfully replicates, with higher anatomical precision, classical findings from fMRI meta-analyses on the localization of temporal processing, while directly linking them with previous electrophysiology and neuroimaging results regarding the underlying neural mechanisms. This work proposes a functional cortical hierarchy for visual temporal processing, where different areas contribute differently to duration perception, as reflected in their neuronal tuning properties. Additionally, it sheds light on the neural basis of the subjective experience of time, suggesting that this may be rooted in the fundamental tuning properties of brain responses.

## Methods

The data presented in this work have been previously used for a different publication [11].

### Participants

Thirteen healthy volunteers participated in this study (6 females; sex information self-reported; gender information not collected; mean age = 29.6, SD = 7.3; 2 left-handed participants). All volunteers had normal or corrected-to-normal visual acuity. The experimental procedures were approved by the International School for Advanced Studies (SISSA) ethics committee (protocol number 11773) in accordance with the Declaration of Helsinki. All participants gave their written informed consent to participate in the experiment, and they were financially compensated for their time and travel expenses.

## Stimuli and procedure

**Stimuli.** Participants were presented with visual stimuli displayed on a BOLD screen (Cambridge Research Systems 32-inch LCD widescreen, resolution = 1920 × 1080 pixels, refresh rate = 120 Hz) placed at a total viewing distance of 210 cm outside the scanner bore and viewed via a mirror. The stimuli were colored circular patches of Gaussian noise subtending 1.5° of visual angle, changing dynamically frame by frame, and presented on a gray background. Each stimulus was constructed by randomly selecting RGB values from a Gaussian distribution of mean = 127 and SD = 35 for each of its pixels and frames. This ensured that the average stimulus luminance was constant and independent of its duration. To prevent the perception of flickers induced by the fast-changing rate in the stimulus, we scaled down its pixel resolution (scaling factor = 12.33). This created a blurring effect that homogenized the local contrasts of the stimulus over frames and minimized possible flickering effects [79]. The entire experimental procedure was generated and delivered using MATLAB and Psychtoolbox-3 [80]. An identical set-up was used during participants training.

**Task and experimental design.** Participants were asked to perform a categorization task. They had to compare the duration (i.e., display time) of a comparison stimulus to the duration of a reference stimulus internalized during the training procedure. The task was to report whether the comparison stimulus was longer or shorter than the reference. The reference duration was 0.5 s, and the comparison durations were 0.2, 0.3, 0.4, 0.6, 0.7, and 0.8 s. The comparisons were presented in different spatial positions (i.e., display locations on the screen), which could be either at 0.9° or 2.5° of visual angle diagonally from the center of the screen in the lower-left or lower-right visual quadrant (see Fig 1A). Stimuli did not overlap across spatial positions. A white fixation cross (0.32° of visual angle) was displayed at the center of the screen throughout the experiment. In each trial, the comparison stimulus was presented in a specific position and entailed a specific duration. After a randomized interval from the offset of the stimulus (stimulus-cue interval (SCI) uniformly distributed between 0.9 and 1.2 s), the participants' response was cued with a color switch (from white to black) in the fixation cross. The response was allowed within a 2 s window, but no emphasis was placed on reaction times. Participants were instructed to provide their responses by pressing one of two buttons on a response pad with their right index finger or right middle finger to express the choices "comparison longer than reference" and "comparison shorter than reference," respectively. No feedback was provided after the response. A uniformly distributed inter-trial interval (ITI) between 1.8 and 2.5 s interleaved the trials. See Fig 1A for a pictorial representation of the trial structure. The stimulus duration varied trial by trial in a pseudo-randomized and counterbalanced fashion, whereas its position varied sequentially and cyclically to minimize attentional switching effects on the duration judgment [81]. Each cycle started and ended at 2.5° in the lower-left quadrant and comprised a clockwise and counterclockwise presentation of the stimulus in all positions, from 2.5° lower-left to 2.5° lower-right and back (i.e., 2.5° L, 0.9° L, 0.9° R, 2.5° R, 2.5° R, 0.9° R, 0.9° L, 2.5° L). Each half cycle (i.e., when the presentation order turned from clockwise to counterclockwise and vice versa) was followed by a 2.64 s (2 TR) interval. To ensure a balanced presentation of all combinations of durations and positions within each block, a cycle was repeated 6 times, and each duration was presented twice in each position, for a total of 48 trials per block. Each participant performed 10 blocks inside the scanner acquired in separate fMRI runs. Duration randomization differed in each block, whereas the position sequence was always the same. Stimuli presentation was synchronized with the scanner acquisition at the beginning and at the middle of each cycle. Participants were instructed to maintain their gaze at the fixation cross while performing the task, and eye movements were recorded with an MR-compatible eye-tracking system (R Research Eyelink 1000 Plus) placed inside the scanner bore.

**Training.** Participants underwent a training procedure outside the scanner to familiarize themselves with the stimuli and the task. First, they were asked to internalize the duration of the reference stimulus. In this phase, participants passively viewed a 0.5 s stimulus presented at the center of the screen 3 times (inter-stimulus interval uniformly distributed between 1.8 and 2.5 s) in each trial. They were free to complete as many trials as they needed to feel confident they had internalized the duration of the stimulus. Next, participants performed the first training block of the duration

categorization task. The task structure was identical as described before (see *Methods - Stimuli and procedure - Task and experimental design*), but all comparisons were presented at the center of the screen. This was done to ensure that participants were able to correctly discriminate the comparisons from the reference stimulus. Finally, participants performed a second training block identical to the experimental blocks to familiarize themselves with the experimental procedure. Throughout the training phase, participants received visual feedback about their performance and eye movements.

## MRI acquisition

MRI data were acquired with a Philips Achieva 7T scanner equipped with an 8Tx/32Rx-channel Nova Medical head coil. T2*-weighted functional images were acquired using a three-dimensional EPI sequence with anterior-posterior phase encoding direction and the following parameters: voxel resolution = 1.8 mm isometric; repetition time (TR) = 1.32 s; echo time (TE) = 0.017 s; flip angle = 13 degrees; bandwidth = 1750 Hz/px. Universal kt-points pulses were used to achieve a more homogeneous flip angle throughout the brain [82]. The matrix size was 112×112×98, resulting in a field of view of 200(AP) × 200(FH) × 176.4(LR) mm. At the end of each run, 4 volumes were acquired with the opposite phase encoding direction in order to perform susceptibility distortion correction (see *Methods - MRI data preprocessing*). A minimum of 190 volumes (acquisition time ≈ 4 min) was acquired for each experimental run. Peripheral pulse and respiratory signals were recorded simultaneously with the fMRI data acquisition using the Philips MR Physiology recording system. The finger clip of the peripheral pulse unit was placed on the subject's left ring finger, and the respiratory sensor was placed over the diaphragm and secured with a band. Eye movements were monitored and recorded with an eye-tracking system (SR Research Eyelink 1000 Plus) mounted onto a hot-mirror system and located inside the scanner bore. High-resolution T1-weighted images were obtained using the MP2RAGE pulse sequence [83] optimized for 7T (voxel size = 0.7 × 0.7 × 0.7 mm, matrix size = 352×352×263).

## MRI data preprocessing

Pulse-oximetry and respiratory components were regressed out from functional MRI data before the preprocessing. We converted physiological signals into slice-based regressors using RetroTS.py (AFNI), and we performed the Retrospective Image Correction with a custom routine based on 3dretroicor (AFNI). This procedure was applied only when physiological signals showed a reliable frequency spectrum (they were removed in 100 out of 130 runs). MRI data were preprocessed using fMRIPrep (v21.0.2). The full description of the preprocessing workflow, automatically generated by fMRIPrep, is provided below.

Results included in this manuscript come from preprocessing performed using *fMRIPrep* 21.0.2 ([84]; [85]; RRID:SCR_016216), which is based on *Nipype* 1.6.1 ([86]; [87]; RRID:SCR_002502).

**Preprocessing of B0 inhomogeneity mappings.** A total of 12 fieldmaps were found available within the input BIDS structure for each subject. A *B0*-nonuniformity map (or *fieldmap*) was estimated based on two (or more) echo-planar imaging (EPI) references with topup ([88]; FSL 6.0.5.1:57b01774).

**Anatomical data preprocessing.** A total of 1 T1-weighted (T1w) images were found within the input BIDS dataset. The T1-weighted (T1w) image was corrected for intensity non-uniformity (INU) with N4BiasFieldCorrection [89], distributed with ANTs 2.3.3 [[90], RRID:SCR_004757], and used as T1w-reference throughout the workflow. The T1w-reference was then skull-stripped with a *Nipype* implementation of the antsBrainExtraction.sh workflow (from ANTs), using OASIS30ANTs as target template. Brain tissue segmentation of cerebrospinal fluid (CSF), white matter (WM), and gray matter (GM) was performed on the brain-extracted T1w using fast [[91], FSL 6.0.5.1:57b01774, RRID:SCR_002823,]. Brain surfaces were reconstructed using recon-all [[92], FreeSurfer 6.0.1, RRID:SCR_001847], and the brain mask estimated previously was refined with a custom variation of the method to reconcile ANTs-derived and FreeSurfer-derived segmentations of the cortical gray matter of Mindboggle [93], RRID:SCR_002438]. Volume-based spatial normalization to two standard

spaces (MNI152NLin6Asym, MNI152NLin2009cAsym) was performed through nonlinear registration with antsRegistration (ANTs 2.3.3), using brain-extracted versions of both T1w reference and the T1w template. The following templates were selected for spatial normalization: *FSL's MNI ICBM 152 non-linear 6th Generation Asymmetric Average Brain Stereotaxic Registration Model* [[94], RRID:SCR_002823; TemplateFlow ID: MNI152NLin6Asym], *ICBM 152 Nonlinear Asymmetrical template version 2009c* [[95], RRID:SCR_008796; TemplateFlow ID: MNI152NLin2009cAsym].

**Functional data preprocessing.** For each of the 12 BOLD runs found per subject (across all tasks and sessions), the following preprocessing was performed. First, a reference volume and its skull-stripped version were generated using a custom methodology of *fMRIPrep*. Head-motion parameters with respect to the BOLD reference (transformation matrices, and six corresponding rotation and translation parameters) are estimated before any spatiotemporal filtering using mcflirt [[96], FSL 6.0.5.1:57b01774]. The estimated *fieldmap* was then aligned with rigid-registration to the target EPI (echo-planar imaging) reference run. The field coefficients were mapped on to the reference EPI using the transform. The BOLD reference was then co-registered to the T1w reference using bbregister (FreeSurfer) which implements boundary-based registration [97]. Co-registration was configured with six degrees of freedom. Several confounding time-series were calculated based on the *preprocessed BOLD*: framewise displacement (FD), DVARS, and three region-wise global signals. FD was computed using two formulations following Power (absolute sum of relative motions, [98]) and Jenkinson (relative root mean square displacement between affines, [96]). FD and DVARS are calculated for each functional run, both using their implementations in *Nipype* following the definitions by [98]. The three global signals are extracted within the CSF, the WM, and the whole-brain masks. Additionally, a set of physiological regressors were extracted to allow for component-based noise correction [[99], *CompCor*]. Principal components are estimated after high-pass filtering the *preprocessed BOLD* time-series (using a discrete cosine filter with 128s cut-off) for the two *CompCor* variants: temporal (tCompCor) and anatomical (aCompCor). tCompCor components are then calculated from the top 2% variable voxels within the brain mask. For aCompCor, three probabilistic masks (CSF, WM, and combined CSF+WM) are generated in anatomical space. The implementation differs from that of Behzadi and colleagues in that instead of eroding the masks by 2 pixels on BOLD space, the aCompCor masks are subtracted a mask of pixels that likely contain a volume fraction of GM. This mask is obtained by dilating a GM mask extracted from the FreeSurfer's *aseg* segmentation, and it ensures components are not extracted from voxels containing a minimal fraction of GM. Finally, these masks are resampled into BOLD space and binarized by thresholding at 0.99 (as in the original implementation). Components are also calculated separately within the WM and CSF masks. For each CompCor decomposition, the *k* components with the largest singular values are retained, such that the retained components' time series are sufficient to explain 50 percent of variance across the nuisance mask (CSF, WM, combined, or temporal). The remaining components are dropped from consideration. The head-motion estimates calculated in the correction step were also placed within the corresponding confounds file. The confound time series derived from head motion estimates and global signals were expanded with the inclusion of temporal derivatives and quadratic terms for each [100]. Frames that exceeded a threshold of 0.5 mm FD or 1.5 standardised DVARS were annotated as motion outliers. The BOLD time series were resampled into standard space, generating a *preprocessed BOLD run in MNI152NLin6Asym space*. First, a reference volume and its skull-stripped version were generated using a custom methodology of *fMRIPrep*. The BOLD time-series were resampled onto the following surfaces (FreeSurfer reconstruction nomenclature): *fsnative*, *fsaverage*. Automatic removal of motion artifacts using independent component analysis [[101], ICA-AROMA] was performed on the *preprocessed BOLD on MNI space* time-series after removal of non-steady state volumes and spatial smoothing with an isotropic, Gaussian kernel of 6 mm FWHM (full-width half-maximum). Corresponding "non-aggresively" denoised runs were produced after such smoothing. Additionally, the "aggressive" noise-regressors were collected and placed in the corresponding confounds file. All resamplings can be performed with *a single interpolation step* by composing all the pertinent transformations (i.e., head-motion transform matrices, susceptibility distortion correction when available, and co-registrations to anatomical and output spaces). Gridded (volumetric) resamplings were performed using antsApplyTransforms (ANTs), configured with Lanczos

interpolation to minimize the smoothing effects of other kernels [102]. Non-gridded (surface) resamplings were performed using *mri_vol2surf* (FreeSurfer).

Many internal operations of *fMRIPrep* use *Nilearn* 0.8.1 [[103], RRID:SCR_001362], mostly within the functional processing workflow. For more details of the pipeline, see the section corresponding to workflows in fMRIPrep'sdocumentation.

**Copyright waiver.** The above boilerplate text was automatically generated by fMRIPrep with the express intention that users should copy and paste this text into their manuscripts *unchanged*. It is released under the CC0 license.

## Behavioral performance analysis

We first assessed that the display location of the stimulus did not bias duration categorization or affect its sensitivity (see [11] for full statistical analysis and results). Next, we estimated individual psychometric curves based on the average fraction of "comparison longer than reference" responses for each comparison duration, collapsing across spatial positions (Fig 7A, gray curves). We also computed the mean fraction of "comparison longer than reference" responses across participants for each comparison duration to estimate the group psychometric curve shown in blue in Fig 7A. All psychometric curves were fitted using the glmfit function in MATLAB with a logit link function. For each participant, we then derived the Point of Subjective Equality (PSE), which represents the comparison duration equally likely to be judged as longer or shorter than the reference and thus quantifies the bias in the duration categorization task. Individual PSE values were used to investigate the link between duration tuning and perception as described in *Methods - Analysis of the link between duration preferences and perception*.

## General linear model (GLM) analysis

We initially analyzed functional MRI data, resampled on the cortical native surface (Freesurfer's fsnative), with a GLM approach using the GLMdenoise toolbox [104]. For each run, the design matrix included one regressor for each combination of comparison duration and position, time-locked to the offset of each stimulus (events of interest), and one regressor time-locked to the onset of each response (event of no interest). In total, we modeled 25 events (6 stimulus durations × 4 stimulus positions + response). As GLMdenoise automatically estimates noise regressors, no motion correction parameters were entered in the procedure. Regressors were convolved with the canonical hemodynamic response function (HRF). For each subject, this procedure yielded a set of 100 bootstrapped beta weights for each vertex. In the subsequent modeling procedure, we used the median beta weights across bootstraps, converted to percent signal change. To perform additional checks on the relationship between brain activity and reaction times (see S13–S15 Tables), we also conducted a single-trial GLM analysis using the GLMsingle toolbox [105]. For each trial, we modeled activity using a single boxcar covering the shortest possible stimulus-cue interval (SCI). For each vertex, this regressor was convolved with the HRF from the GLMsingle library that maximized the GLM fit. Beta estimates were then converted to percent signal change.

## Population receptive field (pRF) modeling

To estimate the tuning properties of BOLD responses elicited by stimulus duration, we modeled the set of GLM beta weights (see *Methods - General linear model (GLM) analysis*) using the pRF approach [28]. We applied a model that assumes a Gaussian response to stimulus duration and invariance to stimulus spatial position. The model is described by the following equation:

$$nr \sim e^{-\frac{(d - \mu_d)^2}{2\sigma_d^2}}$$

(1)

where $\mu_d$ represents the duration preference (i.e., the stimulus duration eliciting the greatest neuronal response) and $\sigma_d$ denotes the sensitivity of the response to changes in stimulus duration (see Fig 1B).

**Fitting procedure.** We first translated our experimental manipulations of the stimulus in a three-dimensional matrix. The stimulus spatial and temporal dimensions were mapped along the first and second axes in arbitrary units ranging from 1 to 100. The third dimension represented all combinations of stimulus duration and position, ordered according to GLM beta weights. To derive the predicted neuronal response, we multiplied the pRF model by the three-dimensional stimulus matrix and then integrated over the first and second dimensions, yielding a series of 24 predictions (i.e., one for each combination of stimulus duration and position). For each vertex of the cortical surface, we optimized this predicted neuronal response by minimizing the residual sum of squares relative to the GLM beta weights. This optimization process was performed in two steps. First, a grid search tested the performance of a large set of parameters for the pRF model. Next, the best-performing parameters from the grid fit were used as seed in an iterative procedure that explored previously untested parameter combinations. The iterative optimization was based on the Nelder-Mead method [106], as implemented in the fminsearchbnd function in MATLAB. Only parameters accounting for at least 10% of variance in the grid fit were further optimized in the iterative step. Vertices displaying a negative pRF were excluded from further analyses. We performed the entire procedure using custom functions in MATLAB. Fig 1B shows the fitting result for a representative vertex.

## Regions of interest (ROIs) identification

We conducted our analyses within a set of ROIs identified using a mass-univariate GLM approach performed with the Statistical Parametric Mapping toolbox in MATLAB (SPM12, version 7219, Wellcome Department of Imaging Neuroscience, University College London). The volumetric functional data, resampled on the MNI152NLin6Asym space, were first smoothed with a 2 mm FWHM Gaussian kernel. For the first-level analysis, we constructed a design matrix that included one regressor for each combination of comparison duration and position, time-locked to the offset of the stimuli (events of interest), and one regressor time-locked to the onset of the responses (event of no interest). All event durations were set to zero, and regressors were convolved with the canonical HRF. The model also included 6 motion correction parameters. For each subject, we then estimated one contrast for each stimulus duration, regardless of spatial position. The resulting contrast images were entered into a second-level full-factorial analysis, where an overall $t$-contrast was computed. The statistical threshold was set at $p < 0.001$, FWE cluster-level corrected for multiple comparisons across the entire brain volume, with cluster size estimated at $p$ FWE-uncorrected$= 0.001$. The volume of group-level clusters was then resampled onto the fsaverage surface using FreeSurfer's mri_vol2surf. To minimize potential statistical inflation due to surface resampling, we applied a minimum $t$-value threshold of 4. Additionally, small clusters were excluded using a minimum surface threshold of 20 mm$^2$.

To identify the anatomical locations of $t$-value clusters, we used the HCP MMP 1.0 atlas [26] in fsaverage space. An atlas area was considered an ROI if at least 5% of its surface was covered by one cluster. For clusters within motor and somatosensory areas, we used the same procedure using the topological atlas by Sereno and colleagues [27] to achieve finer parcellation. This process resulted in the identification of 47 ROIs in the left hemisphere and 46 ROIs in the right, with 29 ROIs shared between hemispheres. Finally, we grouped ROIs into 9 functional streams based on their anatomical locations and their description provided in the multipart supplement of the HCP atlas [107]. All ROIs are listed in Table 1 and displayed in Figs 1C and S1. After identification on the fsaverage surface, the ROIs were subsequently extracted from each participant's native surface for further analyses.

## Analysis of duration preference changes along the cortical hierarchy

In this first analysis, we investigated how the distribution of duration preferences (i.e., the values of $\mu_d$ parameter) changed across different areas. For each participant, we calculated the median $\mu_d$ within each ROI, including both left

**Table 1. Regions of interest (ROIs). ROIs are listed separately for the left and right hemispheres and grouped by functional streams. The ROI nomenclature follows that provided with the atlases [27,26].**

| Functional stream | Left hemisphere ROIs | Right hemisphere ROIs |
|---|---|---|
| Ventral visual (VV) | V4, V8, PIT, FFC, VVC, TE2p | V4, PIT, FFC, VVC, TE2p |
| Lateral visual (LV) | V3CD, LO2, MT, MST, FST, PH | V3CD, LO1, LO2, LO3, V4t, MT, MST, FST, PH |
| Intraparietal sulcus (IPS) | V3B, V7, IP0, IPS1, 7PL, VIP, MIP, LIPv, LIPd, IP2, AIP | V3B, IP0, MIP, LIPv, LIPd, AIP |
| Inferior parietal (IP) | TPOJ2, TPOJ1, PFt, PFop, OP1 | PGp, TPOJ3, TPOJ2, TPOJ1, STV |
| Motor-somatosensory (mot-som) | 1-hand, 3b-hand, 3a-hand, 4-hand, 4-face | |
| Supplementary motor areas (SMA) | 24dd, 24dv, SCEF | 6mp, 6ma, 24dd, 24dv, p24pr, SFL, SCEF, 8BM, p32pr, a32pr, a24pr |
| Premotor (PM) | 6d, 6a, PEF, 6v, 6r | FEF, 55b, PEF, 6v, 6r |
| Anterior insula (AI) | FOP4, FOP5, AVI | FOP4, FOP5, AVI |
| Inferior frontal (IF) | IFJp, 44, p9_46v | IFSp, p9_46v |

and right hemispheres for bilateral ROIs. We then tested two linear mixed effect (LME) models, using the lme4 R package [108], with the following formulas:

To test changes across streams:

$$\mu_d \sim stream + (1|ROI) + (1|subjectID) \tag{2}$$

To test changes across ROIs:

$$\mu_d \sim ROI + (1|subjectID) \tag{3}$$

The variance explained by each LME model was computed using the MuMIn package [109]. The Satterwaite's method [110] implemented in the lmerTest package [111] was used to estimate the degrees of freedom for the LME model ANOVA. Estimated marginal means were compared using the emmeans package [112], which employs the Kenward–Roger's method [113] for degrees of freedom estimation.

### Analysis of duration preference categories along the cortical hierarchy

We further characterized changes in duration preferences along the cortical hierarchy by performing a categorization analysis in conjunction with the local Moran's *I* statistic.

**Categorization of duration preferences.** For each participant, we categorized vertices based on their duration preference into 5 groups: short (0.2−0.32 s), mid-short (0.32−0.44 s), medium (0.44−0.56 s), mid-long (0.56−0.68 s), and long (0.68−0.8 s). We then computed the fraction of vertices in each category for each stream and ROI, averaging across hemispheres for bilateral ones. These data were analyzed using two separate two-way repeated-measure ANOVA, with category and either stream or ROI as within-subject factors, using the anova_test function in R (from the rstatix package [114]). Marginal means were estimated by fitting two separate linear models using the lm function in R, from the stats package [115], with the following formulas: $\sqrt{fractions} \sim stream * Category$ for streams and $\sqrt{fractions} \sim ROI * Category$ for ROIs. Multiple comparisons were performed with the emmeans package [112]. The square-root transformation improved the normality of the data, ensuring more robust contrast estimates.

**Local Moran's *I*.** For each vertex of the cortical surface, we computed the local Moran's *I* statistic [116], which quantifies the correlation between a vertex's duration preference and the average preference of its neighboring vertices, along with an associated significance assessment. This analysis allowed us to identify spatial clusters of significantly associated vertices and classify their spatial relationships into 4 types: high duration preference surrounded by high (high-high, or HH); low surrounded by low (low-low, or LL); high surrounded by low (high-low, or HL); and low surrounded by high (low-high, or LH). For each participant, we computed the local Moran's *I* across the entire cortical surface of each hemisphere separately, centering duration preferences to the mean preference of that hemisphere. The neighborhood structure included the same number of vertices for all ROIs within a given hemisphere and participant, corresponding to one-fourth of the vertices in the smallest ROI of that hemisphere. Overall, the neighborhood structure included between 59 and 135 vertices. This approach allowed us to capture macroscopic patterns of spatial associations of duration preferences and to compare them across the cortex, at the expense of resolving fine-grained spatial patterns within each ROI. Therefore, the observed proportions of spatial association types should not be interpreted in absolute terms, but rather as a byproduct of this methodological choice. To validate the statistic, we used a conditional permutation procedure in which we computed the statistic 999 times. In each iteration, duration preferences of neighboring vertices were randomly sampled from the cortical surface while keeping the value of the current vertex fixed. Vertices with a *p*-value below 0.05 were considered significant. We implemented this statistic with custom MATLAB functions, based on the GeoDa software [117]. Once we obtained the local cluster map for each hemisphere and participant, we computed the fraction of vertices assigned to each spatial association type (i.e., LL, HH, LH, HL) within each ROI. Since HL and LH associations were nearly absent (see S2 Fig), our subsequent statistical analysis focused on the difference in the fraction of vertices between LL and HH. The LL-HH difference was computed for each participant within each ROI and averaged across hemispheres for bilateral ROIs. We then analyzed these data using the following LME models:

To test changes across streams:

$$LL - HH \sim stream + (1|ROI) + (1|subjectID) \tag{4}$$

To test changes across ROIs:

$$LL - HH \sim ROI + (1|subjectID) \tag{5}$$

This analysis was performed using the same R functions and packages described in *Methods - Analysis of duration preference changes along the cortical hierarchy*. Finally, to examine the overall trend of LL − HH differences across the cortical hierarchy, we averaged them across participants within each ROI and fitted a linear regression line (using the fitlm function in MATLAB) to the data ordered from occipital to frontal regions, as shown in Fig 4B and 4D. For changes across streams (panel B), LL − HH differences were also averaged across ROIs belonging to the same functional stream. We then extracted the slope coefficient with its associated *t*-statistics. To further validate the previous analyses, we also tested the LL and HH fraction values. We conducted two separate two-way repeated-measure ANOVA, with association type (i.e., LL and HH) and either stream or ROI as within-subject factors. Marginal means for multiple comparisons were estimated by fitting two separate linear models with the following formulas: *fractions ∼ stream * Type* for streams and *fractions ∼ ROI * Type* for ROIs. This analysis was performed using the same R functions and packages described in *Methods - Analysis of duration preference categories along the cortical hierarchy - Categorization of duration preferences*. We verified that the estimated marginal means and contrasts were numerically identical to those obtained from LME models including a random intercept for each subject, which produced a singular fit in some cases. The results are reported in S18–S23 Tables.

**Analysis of the topographic organization of duration preferences along the cortical hierarchy**

We assessed the topographic properties of duration preferences across the cortical hierarchy with two spatial statistics methods: the global Moran's *I* statistic and the experimental variogram.

**Global Moran's *I*.** The global Moran's *I* statistic assesses the presence of clustering within a sample by quantifying its overall spatial autocorrelation. It is defined as the average of all local Moran's *I* statistics (i.e., the correlation between each vertex's duration preference and the average preference of its neighbors). A Moran's *I* value close to 0 indicates a random spatial distribution of duration preferences, while values above or below 0 indicate increasing degrees of spatial clustering. A positive Moran's *I* indicates that vertices with similar duration preferences are spatially close, whereas a negative Moran's *I* reflects an inverse relationship, where vertices with high duration preferences are near those with low duration preferences, or vice versa. We computed this statistic separately for each ROI, centering duration preferences to the mean preference of that ROI. The neighborhood structure was defined using the 12 nearest neighboring vertices for each vertex. To estimate *p*-values, we performed a random permutation test, shuffling preferred durations across vertices and recomputing the statistic 999 times. Because each ROI has a unique neighborhood structure, statistical comparisons of Moran's I values across ROIs are not feasible. To obtain the global Moran's *I* for each functional stream, we averaged the statistic across the corresponding ROIs.

**Variogram.** The experimental variogram quantifies the spatial autocorrelation in a sample by measuring how the variance between pairs of vertices changes as a function of their distance. For each ROI, we constructed the variogram by first computing the pairwise distances between all vertices within the region. These distances were then grouped into 2 mm-resolution bins, and for each bin we calculated the variance in duration preferences across vertices. This approach ensured that variance estimation remained independent of the extent of the ROI. Duration preferences of each ROI were z-scored prior to variogram computation. We implemented this procedure using custom MATLAB functions based on the GeostatsPy package [118]. From each variogram, we then extracted two parameters, the nugget and the range. The nugget represents the variance at the shortest distance between vertices (see Figs 5B and S4A). The range corresponds to the distance at which the total variance of the ROI, calculated without accounting for spatial structure, is first reached (see Figs 5C and S5A). We then studied how nuggets and ranges varied across both functional streams and ROIs using the following LME models:

To test changes across streams:

$$\text{nuggets} \sim \text{stream} + (1|\text{subjectID}) \tag{6}$$

$$\sqrt{\text{ranges}} \sim \text{stream} + (1|\text{subjectID}) \tag{7}$$

For this analysis, nuggets and ranges were averaged across hemispheres and ROIs within each functional stream.
To test changes across ROIs:

$$\sqrt{\text{nuggets}} \sim \text{ROI} + (1|\text{subjectID}) \tag{8}$$

$$\log(\text{ranges}) \sim \text{ROI} + (1|\text{subjectID}) \tag{9}$$

For this analysis, nuggets and ranges were averaged across hemispheres for bilateral ROIs.
These analyses were performed using the same R functions and packages described in *Methods - Analysis of duration preference changes along the cortical hierarchy*. Square-root and log transformations were used to improve normality in the sample data and ensure more robust model performance.

## Analysis of long-range relationships between duration preferences

We investigated whether and how changes in duration preferences are related along the cortical hierarchy by performing a correlation analysis across areas. For each participant, we computed the median duration preference within each ROI, including both hemispheres for bilateral ROIs. We then calculated the Kendall's τ correlation matrix across areas using the cor function from the R stats package [115]. We treated this matrix as an adjacency matrix (see S6 Fig) and characterized the resulting network by computing, for each node, its degree, centrality, hubness, and authority using Gephi [119]. In addition, we applied the Louvain community detection algorithm [120] to understand the structure of relationships among the ROIs. To gain a more generalized view of these relationships, we computed the Kendall's τ correlation matrix using the median duration preferences of each functional stream per participant, again including both hemispheres where applicable. This correlation matrix was then used to construct a dendrogram. First, we converted the correlation matrix into a distance matrix using the dist function. Hierarchical clustering was then performed with the hclust function (using the default "complete" method), and the dendrogram was generated using the as.dendrogram function (all functions are from the R stats package [115]).

## Analysis of the link between duration preferences and perception

We investigated whether and where a link exists between duration tuning and perception by performing a correlation analysis between duration preferences and PSE values (see *Methods - Behavioral performance analysis*). For each participant, we computed the median duration preference within each functional stream and ROI. We then calculated the Kendall's τ correlation within each stream and ROI between individual duration preferences and PSE values, using the corr function in MATLAB. To examine the trend of this correlation along the cortical hierarchy, Kendall's τ values were transformed to $z$ values using the formula $z(\tau) = \tanh^{-1}\left(\sin\left(\frac{\pi}{2}\tau\right)\right)$[121], ordered from occipital to frontal regions (as shown in Fig 7D), and fitted with a linear regression line using the fitlm function in MATLAB. Finally, we extracted the slope coefficient with its associated $t$-statistics.

## Supporting information

**S1 Fig. Regions of interest (ROIs).** Top: group-level $t$-value clusters ($p < 0.001$ FWE-corrected for multiple comparisons at the cluster level) are displayed color-coded (from red, $t = 4$, to yellow, $t = 10.7$) on a common (fsaverage) flattened cortical surface. Bottom: the 63 ROIs (47 in the left hemisphere and 46 in the right hemisphere) used in the analyses are shown. They encompass the functional activations displayed in the top panel and were extracted from the HCP MMP 1.0 atlas and Sereno's topological atlas (see *Methods - Regions of interest (ROIs) identification)*. ROI labels are displayed in white and follow the nomenclature of the atlases. ROIs are color-coded according to functional streams: green for ventral visual (VV), blue lateral visual (LV), violet for IPS, purple for inferior parietal (IP), red for motor-somatosensory (mot-som), brown for SMA, orange for premotor (PM), ochre for anterior insula (AI), yellow for inferior frontal (IF). Major sulci are displayed as thick semi-transparent white lines, with the following abbreviations: CAS = calcarine sulcus, LOS = lateral occipital sulcus, ITS = inferior temporal sulcus, STS = superior temporal sulcus, IPS = intraparietal sulcus, SF = Sylvian fissure, CS = central sulcus, IFS = inferior frontal sulcus, SFS = superior frontal sulcus, preCS = precentral sulcus. (PDF)

**S2 Fig. Local spatial associations between duration preferences.** Box plots show the group-level distribution ($n = 13$) of the individual fraction of vertices assigned to each type of local spatial association (based on Moran's $I$ statistic) across the different functional streams. Spatial association types are color-coded: gray for non significant (n.s.), yellow for low-low (LL), orange for high-high (HH), pink for high-low (HL), and purple for low-high (LH). Each dot represents the mean fraction across ROIs and hemispheres for each participant. The horizontal black line indicates the median of the distribution, the box shows the interquartile range, and whiskers represent the minimum and maximum values. Streams are

color-coded on the x-axis: green for ventral visual areas (VV), blue for lateral visual areas (LV), violet for intraparietal sulcus (IPS), purple for inferior parietal areas (IP), red for motor and somatosensory areas (mot-som), brown for supplementary motor areas (SMA), orange for premotor areas (PM), ochre for anterior insula (AI), yellow for inferior frontal areas (IF). See *Methods - Analysis of duration preference categories along the cortical hierarchy - Local Moran's I*. Source data are available at the following link: osf.io/2tequ.
(PDF)

**S3 Fig. Group-level distributions of Moran's *I* statistic values.** Each violin plot represents the group-level distribution ($n = 13$) of Moran's *I* statistic values across streams **(A)** and ROIs **(B)**. Both streams and ROIs are ordered from occipital to frontal and from dorsal to ventral. Streams are color-coded as follows: green for VV, blue for LV, violet for IPS, purple for IP, red for mot-som, brown for SMA, orange for PM, ochre for AI, yellow for IF. ROIs are color-coded according to their respective streams. The left side of each violin represents the left hemisphere (darker shades), while the right side represents the right hemisphere (lighter shades). Dots indicate the median of each distribution, while circles correspond to individual data points. Thick lines represent interquartile ranges. The kernel density estimates were computed using a 7% bandwidth. See *Methods - Analysis of the topographic organization of duration preferences along the cortical hierarchy - Global Moran's I*. Source data are available at the following link: osf.io/2tequ.
(PDF)

**S4 Fig. Group-level distributions of nuggets. (A)** An example variogram highlighting the nugget is shown. Each dot represents the variance ($\gamma$) in duration preferences between pairs of vertices at increasing distance (expressed as a fraction of the total extent of the ROI). Dots shading reflects the number of vertex pairs at each distance, with darker shades indicating a higher count. The solid line ($\gamma = 1$) indicates the total variance of the ROI, computed without accounting for spatial structure. The nugget (i.e., the variance at the shortest distance) is highlighted by the dashed line. Panels **(B)** and **(C)** show the group-level distributions ($n = 13$) of variogram nuggets across streams and ROIs, respectively. Nugget values are expressed as a fraction of the total variance of the ROI. Graphical details are the same as in S3 Fig. Asterisks in panel B indicate streams that are statistically different from the others (see S26 Table). See *Methods - Analysis of the topographic organization of duration preferences along the cortical hierarchy - Variogram*. Source data are available at the following link: osf.io/2tequ.
(PDF)

**S5 Fig. Group-level distributions of ranges. (A)** The same example variogram from S4 Fig is shown, with the range (i.e., the distance at which the total variance of the ROI is reached) highlighted by a gray box. Panels **(B)** and **(C)** show the group-level distributions ($n = 13$) of variogram ranges across streams and ROIs, respectively. Range values are expressed as a fraction of the maximum distance between vertices of the ROI. Graphical details are the same as in S3 Fig. Asterisks in panel B indicate streams that are statistically different from the others (see S27 Table). See *Methods - Analysis of the topographic organization of duration preferences along the cortical hierarchy - Variogram*. Source data are available at the following link: osf.io/2tequ.
(PDF)

**S6 Fig. Network representation of duration preferences correlation matrix.** The group-level Kendall's $\tau$ correlations, computed using the median preferred duration of each ROI and participant, are displayed on the cortical surface (fsaverage). To ease visualization, results are merged onto one hemisphere only. Each ROI is represented as a network node, and the color and width of edges connecting nodes indicate the corresponding correlation values. Node color represents the cluster to which each node belongs. Clusters were computed using the Louvain community detection algorithm. The results of this analysis, along with additional network quantification, are available in the OSF repository of this article as Gephi and Gephi Lite GEXF and JSON files. See *Methods - Analysis of long-range relationships between duration preferences*.
(PDF)

**S1 Table. Median duration preference across streams.** Type III ANOVA on LME model estimates with Satterthwaite's method for degrees of freedom.
(CSV)

**S2 Table. Median duration preference across ROIs.** Two-sided $t$ contrasts between ROIs (Bonferroni-corrected) with Kenward–Roger's method for degrees of freedom.
(CSV)

**S3 Table. Median duration preference across streams.** Two-sided $t$ contrasts between streams (Bonferroni-corrected) with Kenward–Roger's method for degrees of freedom.
(CSV)

**S4 Table. Median duration preference across ROIs.** Two-sided t contrasts between ROIs (Bonferroni-corrected) with Kenward–Roger's method for degrees of freedom. Only significant contrasts are reported.
(CSV)

**S5 Table. Categorization of duration preferences across streams.** Type III two-way repeated-measures ANOVA.
(CSV)

**S6 Table. Categorization of duration preferences across streams.** Mauchly's sphericity test.
(CSV)

**S7 Table. Categorization of duration preferences across streams.** Deviation from sphericity correction. GGe = Greenhouse-Geisser ∈, HFe = Huynh-Feldt ∈.
(CSV)

**S8 Table. Categorization of duration preferences across ROIs.** Type III two-way repeated-measures ANOVA.
(CSV)

**S9 Table. Categorization of duration preferences across ROIs.** Mauchly's sphericity test.
(CSV)

**S10 Table. Categorization of duration preferences across ROIs.** Deviation from sphericity correction. GGe = Greenhouse-Geisser ∈, HFe = Huynh-Feldt ∈.
(CSV)

**S11 Table. Categorization of duration preferences across streams.** Two-sided t contrasts between categories for each stream (Bonferroni-corrected). Contrast estimates are expressed as difference of square-root scaled values.
(CSV)

**S12 Table. Categorization of duration preferences across ROIs.** Two-sided t contrasts between categories for each ROI (Bonferroni-corrected). Contrast estimates are expressed as difference of square-root scaled values. Only significant contrasts are reported.
(CSV)

**S13 Table. Reaction times (RTs) across comparison durations.** Type III ANOVA on LME model estimates with Satterthwaite's method for degrees of freedom. The LME model formula is $log(RT) \sim duration + (1|subjectID)$ (marginal $R^2$: 0.01, conditional $R^2$: 0.32). RTs were calculated as the difference between response and cue onsets, and they were log-transformed to improve normality and ensure more robust model performance.
(CSV)

 

**S14 Table. Reaction times (RTs) across comparison durations.** Two-sided t contrasts between comparison durations (Bonferroni-corrected). Contrast estimates are expressed as difference of log-scaled values.
(CSV)

**S15 Table. Brain activity (single-trial β) across reaction times (RTs).** Type III ANOVA on LME model estimates with Satterthwaite's method for degrees of freedom. The LME model formula is $\beta \sim log(RT) + (log(RT)|ID) + (1|subjectID)$ (marginal $R^2$: 0.00055, conditional $R^2$: 0.102). RTs were log-transformed to improve normality and ensure more robust model performance. For details about single-trial β estimation, see *Methods - General linear model (GLM) analysis*.
(CSV)

**S16 Table. Difference between fractions of vertices showing LL and HH types of spatial association across streams.** Type III ANOVA on LME model estimates with Satterthwaite's method for degrees of freedom.
(CSV)

**S17 Table. Difference between fractions of vertices showing LL and HH types of spatial association across ROIs.** Type III ANOVA on LME model estimates with Satterthwaite's method for degrees of freedom.
(CSV)

**S18 Table. Fractions of vertices showing LL and HH types of spatial association across streams.** Type III two-way repeated-measures ANOVA.
(CSV)

**S19 Table. Fractions of vertices showing LL and HH types of spatial association across streams.** Mauchly's sphericity test.
(CSV)

**S20 Table. Fractions of vertices showing LL and HH types of spatial association across streams.** Deviation from sphericity correction. GGe = Greenhouse-Geisser ϵ, HFe = Huynh-Feldt ϵ.
(CSV)

**S21 Table. Fractions of vertices showing LL and HH types of spatial association across ROIs.** Type III two-way repeated-measures ANOVA. Mauchly's test of sphericity could not be evaluated due to the large number of within-subject levels.
(CSV)

**S22 Table. Fractions of vertices showing LL and HH types of spatial association across streams.** Two-sided *t* contrasts between HH and LL spatial associations for each stream (Bonferroni-corrected).
(CSV)

**S23 Table. Fractions of vertices showing LL and HH types of spatial association across ROIs.** Two-sided t contrasts between HH and LL spatial associations for each ROI (Bonferroni-corrected).
(CSV)

**S24 Table. Nuggets across streams.** Type III ANOVA on LME model estimates with Satterthwaite's method for degrees of freedom.
(CSV)

**S25 Table. Ranges across streams.** Type III ANOVA on LME model estimates with Satterthwaite's method for degrees of freedom.
(CSV)

**S26 Table. Nuggets across streams.** Two-sided $t$ contrasts between streams (Bonferroni-corrected) with Kenward–Roger's method for degrees of freedom.
(CSV)

**S27 Table. Ranges across streams.** Two-sided $t$ contrasts between streams (Bonferroni-corrected) with Kenward–Roger's method for degrees of freedom. Contrast estimates are expressed as difference of square-root scaled values.
(CSV)

**S28 Table. Nuggets across ROIs.** Type III ANOVA on LME model estimates with Satterthwaite's method for degrees of freedom.
(CSV)

**S29 Table. Ranges across ROIs.** Type III ANOVA on LME model estimates with Satterthwaite's method for degrees of freedom.
(CSV)

**S30 Table. Nuggets across ROIs.** Two-sided $t$ contrasts between ROIs (Bonferroni-corrected) with Kenward–Roger's method for degrees of freedom. Contrast estimates are expressed as difference of square-root scaled values. Only significant contrasts are reported.
(CSV)

**S31 Table. Ranges across ROIs.** Two-sided $t$ contrasts between ROIs (Bonferroni-corrected) with Kenward–Roger's method for degrees of freedom. Contrast estimates are expressed as difference of log-scaled values. Only significant contrasts are reported.
(CSV)

## Acknowledgments

We thank the staff of the Spinoza Centre for Neuroimaging (www.spinozacentre.nl) for their help and assistance during data collection.

## Author contributions

**Conceptualization:** Valeria Centanino, Gianfranco Fortunato, Domenica Bueti.

**Data curation:** Valeria Centanino, Gianfranco Fortunato.

**Formal analysis:** Valeria Centanino, Gianfranco Fortunato.

**Funding acquisition:** Domenica Bueti.

**Investigation:** Valeria Centanino, Gianfranco Fortunato.

**Methodology:** Valeria Centanino, Gianfranco Fortunato.

**Project administration:** Domenica Bueti.

**Resources:** Domenica Bueti.

**Software:** Valeria Centanino, Gianfranco Fortunato.

**Supervision:** Domenica Bueti.

**Validation:** Valeria Centanino, Gianfranco Fortunato.

**Visualization:** Valeria Centanino, Gianfranco Fortunato.

**Writing – original draft:** Valeria Centanino, Gianfranco Fortunato.

**Writing – review & editing:** Valeria Centanino, Gianfranco Fortunato, Domenica Bueti.

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
