## [Editor Report · Decision Letter 0]

4 Aug 2025

Dear Dr Centanino,

Thank you for submitting your manuscript entitled "Defining a functional hierarchy of millisecond time: from visual stimulus processing to duration perception" for consideration as a Research Article by PLOS Biology.

Your manuscript has now been evaluated by the PLOS Biology editorial staff and I am writing to let you know that we would like to send your submission out for external peer review.

Once your full submission is complete, your paper will undergo a series of checks in preparation for peer review. After your manuscript has passed the checks it will be sent out for review. To provide the metadata for your submission, please Login to Editorial Manager (https://www.editorialmanager.com/pbiology) within two working days, i.e. by Aug 06 2025 11:59PM.

Kind regards,

Christian

Christian Schnell, PhD

Senior Editor

PLOS Biology

cschnell@plos.org

---

## [Decision Letter · Decision Letter 1]

7 Oct 2025

Dear Dr Centanino,

Thank you for your patience while your manuscript "Defining a functional hierarchy of millisecond time: from visual stimulus processing to duration perception" was peer-reviewed at PLOS Biology. First of all, please allow me to apologize for the long delay in sending our decision. One of the reviewers was unfortunately unable to submit their report. In any case, your manuscript has now been evaluated by the PLOS Biology editors, an Academic Editor with relevant expertise, and by several independent reviewers.

In light of the reviews, which you will find at the end of this email, we would like to invite you to revise the work to thoroughly address the reviewers' reports.

As you will see below, the reviewers overall find your study interesting and well done. However, they both raise a few concerns, that need to be addressed via textual revisions and additional analyses.

Given the extent of revision needed, we cannot make a decision about publication until we have seen the revised manuscript and your response to the reviewers' comments. Your revised manuscript is likely to be sent for further evaluation by all or a subset of the reviewers.

**IMPORTANT - SUBMITTING YOUR REVISION**

*Re-submission Checklist*

*Published Peer Review*

*PLOS Data Policy*

*Blot and Gel Data Policy*

Sincerely,

Christian

Christian Schnell, PhD

Senior Editor

PLOS Biology

cschnell@plos.org

REVIEWS:

Reviewer #1: Centanino, Fortunato and Bueti present a manuscript of a re-analysis of data from a previously published report (Centanino, et al. 2024) in which human subjects (n=13) performed a time discrimination task using visual stimuli at distinct spatial locations and sub-second stimuli while being scanned at 7T. While the previous study focused on the dissociation between spatial position and temporal processing, this one instead is focused on tuning properties for durations, as an extension of previous work from this lab on gradient activation. The authors take a population receptive field modeling approach, in which the activity of vertices is analyzed by comparing to a Gaussian distribution at each of the presented durations to determine the maximally-responsive duration for each vertex, thereby determining which regions are most active to a particular interval or interval range. Across a series of analyses, the authors demonstrate that duration tuning appears to progress from early visual areas to parietal, somato-motor and frontal regions in a long-to-short pattern, with posterior regions favoring longer intervals and anterior regions favoring shorter ones, with midline regions such as the SMA and insula favoring the middle of the duration range tested. The authors examine this in various ways using various methods to examine the robustness of their findings, with a final link to behavior by demonstrating that biases in responses can be linked to activity in premotor, SMA, insula, and parietal regions. The authors conclude that early visual areas are monotonically linked to duration encoding, whereas midline areas such as the SMA are linked to the categorical boundary or middle interval and prefrontal regions to response preparation.

Overall, the findings provide a rich and diverse set of data with some very interesting findings. The authors have taken great care in the preparation, curation, and analysis of this dataset, and it offers a nice complement to their previous paper in 2024. There are, however, some questions that I have regarding the findings as well as the presentation that I think would strengthen the manuscript, or at least provide some clarity to readers going forward.

MAJOR:

-The analyses are fairly complicated and so some effort at "unpacking" is helpful here. I think the authors have done a good job but the presentation can get muddled at times. In particular, I found Figure 2 confusing. The authors note that these are maps of duration preferences within each given region, from which I interpret the dominant blue shadings in early visual areas and red shadings in frontal areas to reflect the mean duration preferences for these regions as displayed in Figure 3, yet what do the individual pixels represent? That is, what are the x and y axes here? Is each region flattened/reduced to the same size? Why are some pixels white for some subjects and not others?

-The local Moran's statistics is interesting, but I'm not sure why there would be a lack of HH and LH associations. If the statistic looks for vertices that are completely surrounded by one type of duration tuning them perhaps a lack of LH isn't surprising (a low center surrounded by high duration preferences), but why would there not be any HH associations? I would expect these to be found in visual areas as found using other measures.

-The link to the behavioral data is the most interesting to me, as it provides a nice connection to how these areas are associated with perceptual experience. However some questions remain: first, did the authors explore any other behavioral measures here? The coefficient of variation (CV) could be just as informative as the PSE here, by demonstrating how the variance or "noisiness" of the representation is affected by the tuning properties of individual regions/subjects. Second, if the prefrontal regions are indeed reflecting motor preparation then one would expect a link here with reaction time (RT). I recognize that the task was not speeded, yet the authors could see if 1) RTs nonetheless differed as a function of duration, and 2) activity in the frontal regions correlates with the speed of response, thus supporting their argument.

-Connected to the above, the authors don't go into any detail regarding the negative correlations that are observed here. That is, there appears to be a negative correlation between VVC and V3CD and PSE, such that higher activity in these regions corresponds to shorter perceived durations, which seems opposite to the argument the authors make that activity in these regions reflects monotonic increases with longer durations.

MINOR:

-Figure 4 is a little confusing in one regard: why are the lines curved? For example, in panel a there is an "arc" from VV to LV on the blue curve, but why would this be? Shouldn't it be a straight line?

-Given that so many correlations were run, both in Figure 6 and 7, were there any corrections for multiple comparisons?

Reviewer #2: This study investigates the functional imaging topography of preferred durations along the cortical hierarchy during an interval categorization task. The results showed a large representation of long durations in visual cortex, whereas there is a bias for shorter duration preferences in motor and somatosensory cortices. In addition, all presented durations were topographically organized as interval tuned clusters in parietal and premotor cortices, as well as in the caudal supplementary motor area (SMA). Notably, the rostral SMA, inferior frontal cortex, and anterior insula showed neuronal preferences centered around the mean duration, which correlated with the subjective boundary duration participants employed in the task. These findings support the notion of a hierarchical organization of duration tuning spanning from sensory to perceptual interval processing.

This is a well performed and well analyzed and written study providing novel information regarding the hierarchical organization of cortical duration tuning. Crucially, the authors found distinct functional stages of duration processing, which were implemented in specific cortical regions, that were organized hierarchically, and that supported different functions. Nevertheless, I have a series of main concerns. First, the authors used a categorization task with a memorized bisection point not a discrimination task. Please correct. This is important because there is a large literature on the psychophysics of timing with versions of the categorization task that used training of the prototypes, or as in this case, using training of the bisection point. Second, Since the anterior and posterior sections of SMA have different properties on their topography of preferred durations, I suggest to explicitly dividing these two portions across all the figures and analysis, which will make the story clearer. In fact, there is also a bast literature using the subdivision of SMAproper and preSMA, especially in non-human primates. Third, I suggest comparing the categorical behavior of the subjects with the BOLD signal during the SCI. Thus, psychometric decisions can be compared with the choice probability calculated from the betas weights of the model for correct and incorrect decisions for each duration and each subject. This analysis could perfectly complement the PSE analysis on the duration tuning distributions. Finally, the duration tuning without a response modulation for spatial location suggests that the reported timing visual areas are high in the visual processing hierarchy. Nevertheless, the fact that they showed a large bias towards long durations may be an indication of sensory adaptation on short durations instead of accumulation. Indeed, ramping activity is usually not reported in primary sensory areas. Please discuss.

Minor comments.

Please use large fonts in the figure.

Please consider the change of contrast of the ROIS with respect to the inflated brain of Figure 1C, it is difficult to distinguish their limits.

I strongly suggest using graph analysis on the matrix data of Figure 6.

Please consider referring to the work of Merchant et al., 2024 Adv Exp Med Biol regarding the core timing network.

It would be great to run subjects in an experiment with two blocks of intervals to determine how the range of intervals and the position of the bisection point modulate their functional tuning distributions, their topographic organization, and their relationship with the psychometric functions (PSE and choice probability). I know that this implies running a new experiment, but it would be nice to at least discuss the advantages of this experimental design.

---

## [Decision Letter · Decision Letter 2]

11 Feb 2026

Dear Dr Centanino,

Thank you for your patience while we considered your revised manuscript "Defining a functional hierarchy of millisecond time: from visual stimulus processing to duration perception" for publication as a Research Article at PLOS Biology. This revised version of your manuscript has been evaluated by the PLOS Biology editors, the Academic Editor and the original reviewers.

Based on the reviews and on our Academic Editor's assessment of your revision, we are likely to accept this manuscript for publication, provided you satisfactorily address the following data and other policy-related requests:

* We would like to suggest a different title to improve its accessibility for our broad audience:

Neuronal populations across the cortex are differently tuned to discrete, categorical and subjective duration of visual stimuli

* Please add the links to the funding agencies in the Financial Disclosure statement in the manuscript details.

* DATA POLICY:

Regardless of the method selected, please ensure that you provide the individual numerical values that underlie the summary data displayed in the following figure panels as they are essential for readers to assess your analysis and to reproduce it: 1B, 3AB, 4BD, 5ABC, S2, S3AB, S4BC and S5BC.

* CODE POLICY

Per journal policy, if you have generated any custom code during the course of this investigation, please make it available without restrictions. Please ensure that the code is sufficiently well documented and reusable, and that your Data Statement in the Editorial Manager submission system accurately describes where your code can be found. More information on our Code Policy, what and how to share can be found here: https://journals.plos.org/plosbiology/s/code-availability

* If you have any references in the supplementary information, please move them to the main reference list or at least add them there too.

* If you have further methodological details in the supplementary information, please move those to the main manuscript file. We don't have a word count limit and want to make it as easy as possible for readers to find the information they are looking for.

We expect to receive your revised manuscript within two weeks.

*Published Peer Review History*

*Press*

Sincerely,

Christian

Christian Schnell, PhD

Senior Editor

cschnell@plos.org

PLOS Biology

Reviewer remarks:

Reviewer #1: The authors have addressed all of my comments and concerns.

Reviewer #2: The authors did a great job at answering all the reviewers' comments and now the paper delivers an interesting and novel story on temporal processing across the cortical hierarchy.

---

## [Editor Report · Decision Letter 3]

26 Feb 2026

Dear Dr Centanino,

Thank you for the submission of your revised Research Article "Neuronal populations across the cortex underlie discrete, categorical and subjective representations of visual durations" for publication in PLOS Biology. On behalf of my colleagues and the Academic Editor, Christopher Pack, I am pleased to say that we can in principle accept your manuscript for publication, provided you address any remaining formatting and reporting issues. These will be detailed in an email you should receive within 2-3 business days from our colleagues in the journal operations team; no action is required from you until then. Please note that we will not be able to formally accept your manuscript and schedule it for publication until you have completed any requested changes.

PRESS

We have noted your institution's plans about a press release. Please do not opt into the early article posting, so we can schedule the publication of your paper.

Sincerely,

Christian

Christian Schnell, PhD

Senior Editor

PLOS Biology

cschnell@plos.org